# Water-stable boroxine structure with dynamic covalent bonds

Xiaopei Li[1,2], Yongjie Zhang[2], Zhenqiang Shi [ID][1], Dongdong Wang[1], Hang Yang[1], Yahui Zhang [ID][1], Haijuan Qin[3], Wenqi Lu[1], Junjun Chen[1], Yan Li[1] & Guangyan Qing [ID][1,4] ✉

Boroxines are significant structures in the production of covalent organic frameworks, anion receptors, self-healing materials, and others. However, their utilization in aqueous media is a formidable task due to hydrolytic instability. Here we report a water-stable boroxine structure discovered from 2-hydroxyphenylboronic acid. We find that, under ambient environments, 2-hydroxyphenylboronic acid undergoes spontaneous dehydration to form a dimer with dynamic covalent bonds and aggregation-induced enhanced emission activity. Intriguingly, upon exposure to water, the dimer rapidly transforms into a boroxine structure with excellent pH stability and water-compatible dynamic covalent bonds. Building upon these discoveries, we report the strong binding capacity of boroxines toward fluoride ions in aqueous media, and develop a boroxine-based hydrogel with high acid–base stability and reversible gel–sol transition. This discovery of the water-stable boroxine structure breaks the constraint of boroxines not being applicable in aqueous environments, opening a new era of researches in boroxine chemistry.

Boroxines, the dehydration products of organoboronic acids (Fig. 1a), are versatile structures that have found diverse applications in various fields[1–6]. One notable application of boroxines is their use as boron reagents in Suzuki–Miyaura cross-coupling reaction to synthesize valuable products[2,7]. Another area where boroxines show promise is in the development of non-halogenated flame-retardant materials due to their excellent char formation ability and high thermal stability[3,8]. In addition, boroxines have electrophilic properties and can serve as effective anion receptors, making them useful in lithium-ion batteries to improve ionic conductivity[4,9]. Furthermore, the dynamic nature of boron–oxygen (B–O) bonds in boroxines, with the unique combination of kinetic tenability and high thermodynamic stability, has rendered them as promising dynamic covalent motifs for the development of malleable and healable polymers[5,10–12], and this property has also paved the way for covalent organic frameworks (COFs)[13–15].

Despite the diverse range of boroxine utilities across various scientific domains, their functionality in aqueous media can hardly be realized, even under moist conditions[16,17]. That is because boroxines rapidly hydrolyze into boronic acids upon exposure to water (Fig. 1a), resulting in the loss of functionality[18]. Efforts to enhance the stability of boroxines against hydrolysis have traditionally focused on reducing the electrophilicity of the Lewis acidic boron sites. Strategies include introducing electron-donating groups, incorporating bulky groups, and forming adducts with N-donor ligands[18–20]. Recently, Ono et al. proposed an entropic stabilization strategy by incorporating three boronic acid units into a flexible macrocycle[20]. Although these methods have made boroxines more robust against hydrolysis, they have not fundamentally addressed the underlying issue of hydrolytic instability of boroxines. Given the ubiquitous presence of water in the environment, it is of great significance to

[1]CAS Key Laboratory of Separation Science for Analytical Chemistry, Dalian Institute of Chemical Physics, Chinese Academy of Sciences, Dalian, P. R. China. [2]Instrumental Analysis Center, School of Textile and Material Engineering, Dalian Polytechnic University, Dalian, P. R. China. [3]Research Centre of Modern Analytical Technology, Tianjin University of Science & Technology, Tianjin, P. R. China. [4]College of Chemistry and Chemical Engineering, Wuhan Textile University, Wuhan, P. R. China. ✉e-mail: qinggy@dicp.ac.cn

**Fig. 1 | Comparison of phenylboronic acid (PBA) and 2-hydroxyphenylboronic acid (HO-PBA). a** Reversible transformation between PBA and triphenylboroxine (TPB); **b** HO-PBA undergoes spontaneous dehydration to form a dimer under ambient environments, and reversible transformation between HO-PBA dimer and HO-PBA trimer–H$_2$O complex.

address the hydrolytic instability issue of boroxines to expand their applications.

In this study, we discover a boroxine structure that is stable in water, exhibits excellent pH stability, and possesses dynamic covalent bonds (DCBs) compatible with water. This story begins with the study of 2-hydroxyphenylboronic acid (HO-PBA). Through single crystal X-ray diffraction analysis, we find that, under ambient environments, HO-PBA undergoes spontaneous dehydration to form a dimer, which challenges the existing literature's claim of its existence solely as a monomer (Fig. 1b)[21,22]. Further study reveals that the two B–O bonds in HO-PBA dimer are DCBs, allowing for rapid exchange in dry tetrahydrofuran (THF) at room temperature. Besides, the well-ordered stacking of HO-PBA dimers endows it with remarkable aggregation-induced enhanced emission (AIEE) activity, a property, however, typically possessed by aromatic molecules with highly twisted spatial configurations and bulky molecular sizes[23,24]. More interestingly, upon exposure to water, HO-PBA dimers rapidly transform into HO-PBA trimer–H$_2$O complexes at room temperature (Fig. 1b), which contain water-stable boroxine structures that are confirmed through nuclear magnetic resonance (NMR) spectroscopy, mass spectrometry (MS), UV–Raman spectrometry, as well as theoretical calculations. Notably, this water-stable boroxine structure maintains the stability across a broad pH range, and its B–O bonds are dynamic that can exchange rapidly in aqueous media at room temperature. These discoveries lead to the selective recognition of fluoride ions (F⁻) by boroxines in aqueous media, with a significantly stronger affinity than the widely used F⁻ receptor, phenylboronic acid (PBA)[25–27]. Furthermore, we successfully develop a hydrogel cross-linked by the boroxine structures, which demonstrates excellent acid–base stability and reversible gel–sol transition. This work challenges the common knowledge about the hydrolytic instability of boroxines, and provides a promising platform for the construction of dynamic hydrogel, materials for detection and separation of F⁻, hydrophilic COFs, repairable underwater adhesives, materials for biosensing and bioseparation, and others.

## Results

### HO-PBA dimer

HO-PBA is a useful organic molecule that is widely employed in organic reactions and catalysis[22,28,29]. However, its single-crystal structure,

which enables the visualization of intermolecular interactions and provides insights into stereochemistry and reaction selectivity, is currently unavailable. Yet, these pieces of information are crucial for the explanation of the reaction and catalysis mechanisms. Herein, we successfully obtained the single crystal of HO-PBA through slow evaporation of its solution. Surprisingly, the single-crystal structure of HO-PBA challenges the notion in the literature that HO-PBA exists as a monomer[21,22]. Instead, it clearly displays that, under ambient environments, HO-PBA undergoes spontaneous dehydration to form a dimer through the formation of two B–O covalent bonds (Fig. 2a), which was also confirmed by the ¹H and ¹³C NMR spectra (Supplementary Fig. 1a and see below). To make clear the reason behind the instability of HO-PBA monomer under ambient environments, density functional theory (DFT) calculations were performed using M06-2X functional and TZVP basis set. The free energy for the dehydration of HO-PBA monomers to HO-PBA dimers was calculated to be −21.4 kJ mol⁻¹ (Supplementary Table 1), indicating that the instability of HO-PBA monomers under ambient environments can be attributed to the inherent thermodynamic preference for dimerization. Additionally, this dimeric structure was observed in other derivatives of HO-PBA, such as 2-hydroxy-5-methylphenylboronic acid (CH$_3$-HO-PBA) with the electron-donating group −CH$_3$ (Supplementary Figs. 1b, 2a and 13b,c) and 2-hydroxy-4-(trifluoromethyl)phenylboronic acid (CF$_3$-HO-PBA) with the electron-withdrawing group −CF$_3$ (Supplementary Figs. 1c, 3a and 15a, b). This suggests that the formation of dimeric structures might be a common characteristic of HO-PBA and its derivatives.

### DCBs in the HO-PBA dimeric structure

Further investigations unveiled an intriguing aspect regarding the dimeric structure of HO-PBA, wherein the two B–O covalent bonds are DCBs that exhibit reversible breakage and formation properties. This was evidenced by the rapid exchange between different HO-PBA dimeric structures at room temperature. To observe this exchange reaction, HO-PBA dimers and CH$_3$-HO-PBA dimers were co-dissolved in dry THF at room temperature. After evaporation of THF, the resulting powder was characterized using matrix-assisted laser desorption ionization time-of-flight (MALDI-TOF) MS. The mass spectrum shows two peaks at *m/z* 239.18 and 267.22 that correspond to HO-PBA dimer and CH$_3$-HO-PBA dimer, respectively (D0 and D2), and a third peak at

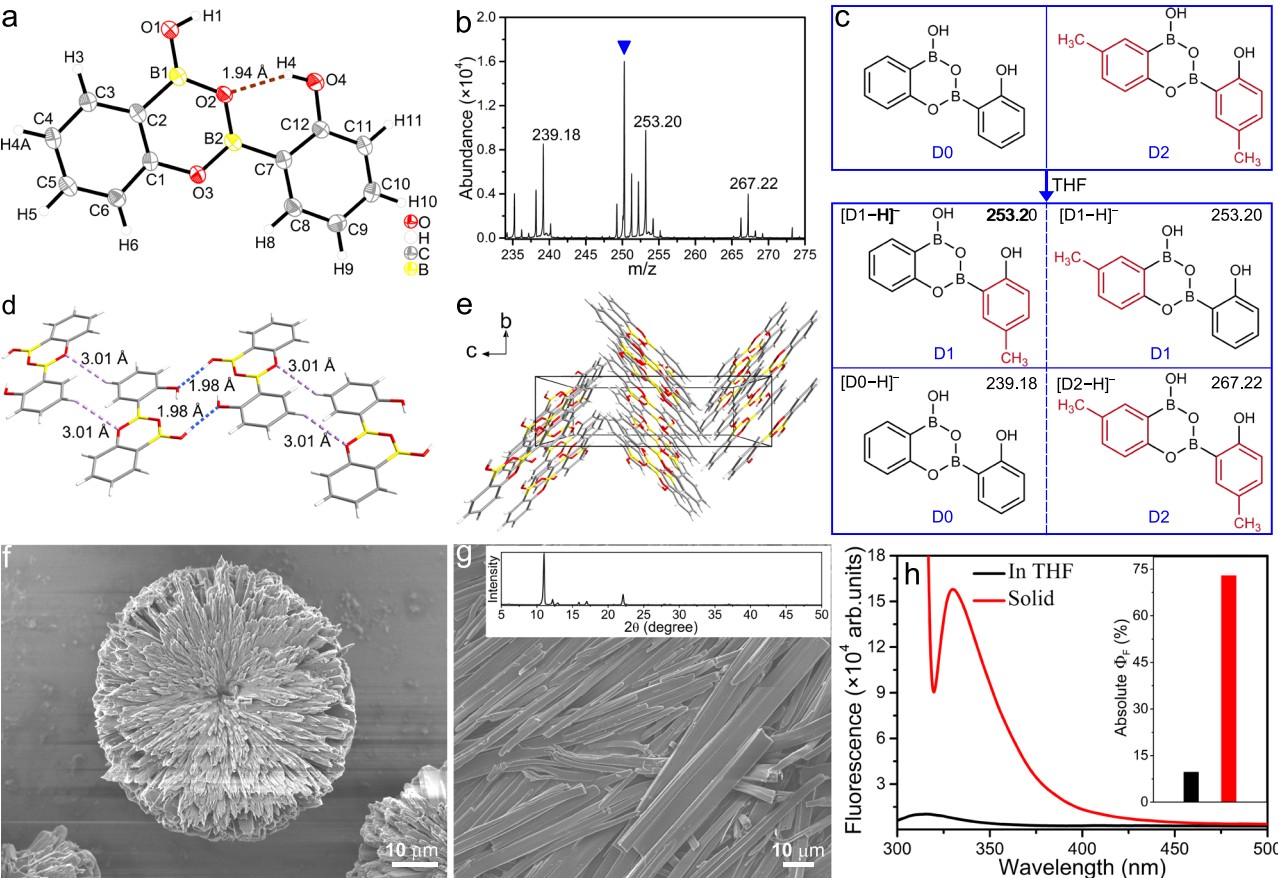

**Fig. 2 | HO-PBA dimer and its properties. a** Crystal structure of HO-PBA dimer with thermal ellipsoids (CCDC 2280484). The dashed brown line represents an intramolecular O–H...O hydrogen bond. **b** MALDI-TOF mass spectrum of products from the mixture of HO-PBA dimers and CH₃-HO-PBA dimers, acquired in a negative mode. The peak marked with a blue triangle is attributed to a background signal. **c** Illustration of the dynamic exchange between HO-PBA dimer (D0) and CH₃-HO-PBA dimer (D2).The exchange products are denoted by D1. **d**, **e** Packing interactions (**d**) and packing mode of HO-PBA dimers viewed along the *a* axis (**e**). The dashed blue and purple lines in (**d**) represent intramolecular O–H...O and C–H...O hydrogen bonds, respectively. **f** SEM image displaying assembled morphology of HO-PBA dimer, where the sample was prepared by dropping one droplet of HO-PBA solution on a silicon wafer. **g** SEM image and XRD pattern (inset) of the solid HO-PBA dimer sample. **h** Fluorescence spectra and the absolute quantum yields ($\Phi_F$, inset) of HO-PBA dimer in THF ($2 \times 10^{-5}$ mol L$^{-1}$, $\lambda_{ex}$ = 288 nm) and the solid HO-PBA dimer sample ($\lambda_{ex}$ = 313 nm).

*m/z* 253.20, attributed to their exchange product (D1) (Fig. 2b, c). This indicated that an exchange reaction occurred between HO-PBA dimer and CH₃-HO-PBA dimer, resulting in a mixture of homo- and hetero-substituted HO-PBA dimeric structure under equilibrium. Similar exchange reactions were also observed between HO-PBA dimer and CF₃-HO-PBA dimer (Supplementary Fig. 4), as well as between CF₃-HO-PBA dimer and CH₃-HO-PBA dimer (Supplementary Fig. 5). DCBs provide a powerful avenue to synthesize complex molecular architectures (e.g., cages, macrocycles and interlocked molecules)[30–32], and materials with advanced properties (e.g., COFs, self-healing materials, and vitrimers)[33,34]. The highly dynamic nature of B–O bonds in the HO-PBA dimeric structure at room temperature gives rise to its tremendous potential in these aspects.

## AIEE property of HO-PBA dimer

In the single-crystal structure, HO-PBA dimers aggregate into an ordered packing by means of intermolecular O–H...O and C–H...O hydrogen bonds (Fig. 2d, e, Supplementary Fig. 6 and Supplementary Tables 2–4). Correspondingly, the scanning electron microscopy (SEM) image of the HO-PBA dimer sample, which was prepared by dropping the HO-PBA solution onto a clean silicon wafer, displays that HO-PBA dimers self-assemble into flower-shaped structures (Fig. 2f). Moreover, the solid HO-PBA dimer sample (Aladdin Reagent, Shanghai, Product No. H101964) has a cuboid shape and exhibits sharp

reflection peaks in its X-ray diffraction (XRD) pattern (Fig. 2g), illustrating its crystalline nature. The highly ordered molecular packing and crystalline structure suggest that the intramolecular motions of HO-PBA dimers might be restricted in the aggregate, leading us to speculate that HO-PBA dimers might exhibit AIEE, a well-known photophysical phenomenon characterized by a stronger emission of molecular aggregates than that of individual molecules[23,24,35]. As shown in Fig. 2h, the solid HO-PBA dimer sample emits strong fluorescence (red line) and possesses an absolute quantum yield ($\Phi_F$) of 73.0%, significantly higher than that of HO-PBA dimer in THF (absolute $\Phi_F$: 9.7%). These fluorescent results confirmed the AIEE property of HO-PBA dimers. Likewise, CH₃-HO-PBA and CF₃-HO-PBA dimers also exhibit AIEE activities (Supplementary Figs. 2, 3, 7–9 and Supplementary Tables 5–10). Typically, conventional aromatic AIEE molecules possess highly twisted spatial configurations and bulky molecular sizes, such as hexaphenylsilole, tetraphenylethene, and their derivatives[23,24]. However, despite its simple molecular structure, HO-PBA dimers display AIEE activity with a remarkably high absolute $\Phi_F$, introducing fresh insights into the AIEE realm.

## Discovery of a water-stable boroxine structure

The AIEE phenomenon is usually exhibited by a gradually enhanced fluorescent emission when the fraction of water (poor solvent) to organic solvent (good solvent, such as THF and methanol) is

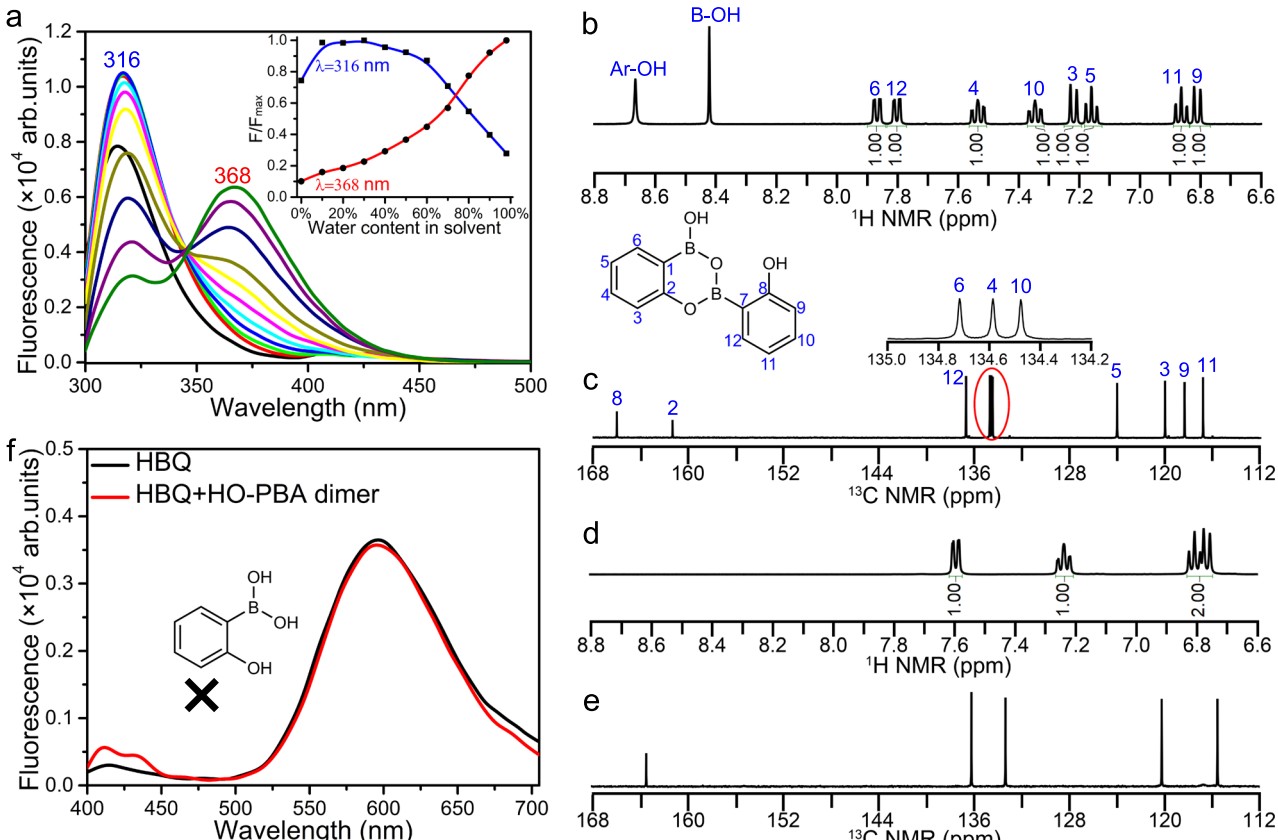

**Fig. 3 | Variations of HO-PBA dimer upon exposure to water. a** Fluorescent spectra and intensity changes (at 316 and 368 nm, inset) of HO-PBA dimer ($2 \times 10^{-5}$ mol $L^{-1}$) in THF–water mixtures with different fractions of water (volume%), $\lambda_{ex} = 288$ nm. **b–e** $^1$H (**b, d**) and $^{13}$C (**c, e**) NMR spectra of HO-PBA dimer in THF–$d_8$ (**b, c**) and in $D_2O$–THF–$d_8$ (2:1, v/v) mixture (**d, e**) at room temperature, concentration: 8 mg $mL^{-1}$. **f** Fluorescent spectrum of 10-hydroxybenzo[h]quinolone (HBQ, $4.0 \times 10^{-5}$ mol $L^{-1}$) before (black) and after (red) addition of five molar equivalents of HO-PBA dimer in a water–THF (4:1, v/v) solution, $\lambda_{ex} = 367$ nm.

increased[24]. However, the HO-PBA dimer's characteristic fluorescent peak at 316 nm in the THF–water solution shows a slow upward trend when the volume fraction of water increases from 0 to 30%, after which it gradually decreases (Fig. 3a). Meanwhile, a new fluorescent peak at 368 nm appears, and its relative intensity ($F/F_{max}$) gradually increases from 0.1 to 1 with the increase of the volume fraction of water (Fig. 3a, the absolute $\Phi_F$ of HO-PBA dimer in water is 19.4%). Similar fluorescent variations of HO-PBA dimer were observed in methanol–water solutions with different water fractions (Supplementary Fig. 10), indicating that the fluorescent variations of HO-PBA dimer are induced by water and independent of the organic solvent employed.

To investigate changes in the HO-PBA dimer solution resulting from the addition of water, NMR characterizations were conducted. The $^1$H (Fig. 3b) and $^{13}$C (Fig. 3c) NMR spectra of HO-PBA dimer in THF–$d_8$ display all the signals of HO-PBA dimer, except for the signals of the two C atoms bonded to B atoms (Fig. 3c, marked with 1 and 7 in the molecular structure of HO-PBA dimer) due to the quadrupolar relaxation mechanism of $^{11}$B nucleus[36]. However, upon addition of water into the THF–$d_8$ solution of HO-PBA dimer, $^1$H (Fig. 3d) and $^{13}$C (Fig. 3e) NMR signals are reduced to four and five sets, respectively, which are consistent with the numbers of H and C present in HO-PBA monomer (The signal of C atom bonded to B atom is too weak to be detected.). Thus, we hypothesized that HO-PBA dimer hydrolyzed into HO-PBA monomer in the presence of water. To test this hypothesis, 10-hydroxybenzo[h]quinolone (HBQ), an effective reagent for detecting boronic acid groups, was employed[37]. Excited HBQ undergoes an excited-state intramolecular proton transfer (ESIPT) process with the maximum emission at 597 nm. Binding with boronic acid groups would

interrupt the ESIPT of HBQ, resulting in a shift of the maximum emission to 504 nm[37]. After mixing HBQ with excess HO-PBA dimers in the solution, the emission peak at 504 nm did not appear (Fig. 3f), which implies no boronic acid groups are present in the water–THF solution of HO-PBA dimer. Therefore, our hypothesis about the transformation of HO-PBA dimer into monomer in the presence of water is incorrect.

The changes in the HO-PBA dimer's solution subsequent to the addition of water were further examined using electrospray ionization quadrupole time-of-flight (ESI-Q-TOF) MS. In Fig. 4a, two strong peaks at $m/z$ 359.13 and 373.14 captured our attention. The $m/z$ 359.13 corresponds to the molecular weight of the deprotonated HO-PBA trimer ([T − H]$^-$), while the $m/z$ 373.14 corresponds to the molecular weight of the deprotonated product [T + CH$_2$ − H]$^-$ from the HO-PBA trimer's etherification with the solvent methanol during the MS measurement. This indicates that HO-PBA dimer might transform into HO-PBA trimer in the presence of water. The $^1$H and $^{13}$C NMR spectra of HO-PBA trimer display four and five sets of signals (Fig. 3d, e), respectively, suggesting that HO-PBA trimer has a symmetric structure. Based on these findings, we boldly speculated that HO-PBA trimer was a molecule with a boroxine structure, which was stable in aqueous media, as depicted in Fig. 4a.

**Characterization of the molecular structure of HO-PBA trimer**

Firstly, the presence of phenolic hydroxyl groups in the molecular structure of HO-PBA trimer was examined by the typical reaction between phenolic hydroxyl groups with $Br_2$ or $Fe^{3+}$. Upon mixing the HO-PBA trimer solution with bromine water, a white precipitate was

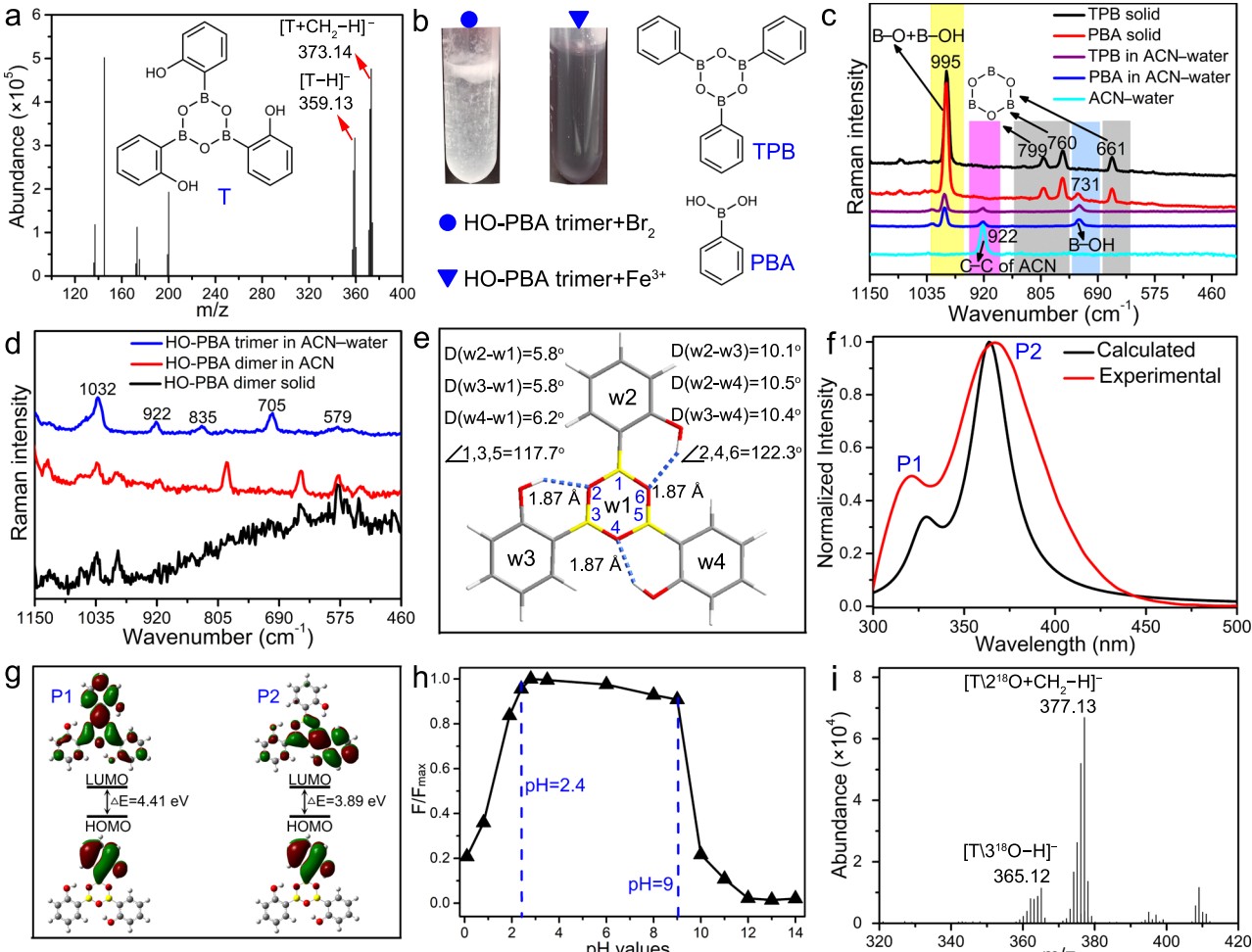

**Fig. 4 | HO-PBA trimer and its structural characterization. a** ESI-Q-TOF mass spectrum of HO-PBA dimer in a methanol−water (5:1, v/v) solution, acquired in a negative mode. HO-PBA trimer is donated by T. **b** Photos of the HO-PBA trimer solution after addition of Br₂ (circle) and FeCl₃ (triangle). **c** UV−Raman spectra of the solid TPB (black) and PBA (red) sample, TPB (purple) and PBA (blue) in an ACN−water (1:2, v/v) solution, and an ACN−water (1:2, v/v) mixture (cyan). **d** UV−Raman spectra of the solid HO-PBA dimer sample (black), HO-PBA dimer in a dry ACN solution (red) and HO-PBA trimer in an ACN−water (1:2, v/v) solution (blue). **e** Optimized geometry of HO-PBA trimer, calculated by Gaussian at the B3LYP/6-31 G* level. The dashed dark blue lines represent intramolecular O−H...O hydrogen bonds. D(w2-w1) represents the dihedral angle between w1 and w2. **f** Comparison of the calculated and experimental fluorescent spectra of HO-PBA trimer in water. **g** Frontier molecular orbitals of the highest occupied molecular orbital (HOMO) and lowest unoccupied molecular orbital (LUMO) of HO-PBA trimer in the excited state. **h** Relative fluorescence intensity at 368 nm of HO-PBA trimer (F/F$_{max}$) as a function of pH value. **i** ESI-Q-TOF mass spectrum of HO-PBA trimer in methanol−H₂¹⁸O (5:1, v/v) solution, acquired in a negative mode.

formed (Fig. 4b). Besides, the addition of freshly prepared ferric chloride solution turned the initially colorless HO-PBA trimer solution into purple (Fig. 4b). These results prove that the phenolic hydroxyl groups are preserved in the molecular structure of HO-PBA trimer.

Secondly, UV−Raman spectrometry was employed to investigate whether a boroxine ring is present in the molecular structure of HO-PBA trimer. For comparison, the UV−Raman spectra of TPB and PBA were also measured. The solid TPB and PBA sample share four UV−Raman peaks at 995, 799, 760, and 661 cm⁻¹ (Fig. 4c, black and red line). The peak at 995 cm⁻¹ is attributed to the stretching vibration of B−O groups and the bending vibration of B−OH groups[38,39]. Due to the ease of dehydration, PBA readily transforms into TPB, resulting in the coexistence of PBA and TPB in the solid PBA sample. Thus, the remaining three peaks could be reasonably assigned to the characteristic vibration of the boroxine ring in TPB. When exposed to water, TPB hydrolyzes into PBA, and the three peaks disappear (Fig. 4c, purple and blue line). This result further supported the assignment of the three peaks to the characteristic vibration of the boroxine ring.

With the characteristic peaks of boroxine ring being confirmed, we focused our attention on the UV−Raman spectrum of HO-PBA

dimer and trimer samples. As shown in Fig. 4d, the UV−Raman spectrum of HO-PBA trimer (blue line) is significantly different from that of HO-PBA dimer in the solid state (black line) and in a dry acetonitrile (ACN) solution (red line). It exhibits four characteristic peaks of the boroxine ring at 1032, 835, 705, and 579 cm⁻¹, respectively, along with a peak at 922 cm⁻¹ assigned to the C-C stretching of ACN (Fig. 4c, cyan line). The discrepancies in peak positions between HO-PBA trimer and the solid TPB sample in the UV−Raman spectra could be attributed to the influence of the phenolic hydroxyl substituents on the skeletal vibration of HO-PBA trimer. The above results confirmed the presence of a boroxine ring within the molecular structure of HO-PBA trimer.

Lastly, DFT calculations further supported the HO-PBA trimer's structure. As displayed in Fig. 4e, the central boroxine skeleton adopts a hexagonal shape that is nearly flat and regular with bond angles close to 120°. The arrangement of the three phenyl rings that are connected to B atoms is found to be non-planar relative to the boroxine ring. Furthermore, intramolecular hydrogen bonding is observed between the boroxine ring and the phenolic hydroxyl groups. Based on the optimized geometry, the NMR and fluorescent spectrum of HO-PBA trimer in water were calculated. The calculated ¹H and ¹³C NMR spectra

of HO-PBA trimer are in agreement with the experimental data (Supplementary Fig. 11). The calculated fluorescent spectrum displays two bands (P1 and P2) located at 328 and 364 nm, respectively, which is also consistent with the experimental data (Fig. 4f). The P1 fluorescence is characteristic of local-excited emission, as determined by the fact that the highest occupied molecular orbital (HOMO) and the lowest unoccupied molecular orbital (LUMO) of P1 are mainly localized on the same phenyl fragment (Fig. 4g). The HOMO and LUMO of P2 are distributed on the different fragments of HO-PBA trimer, indicating that the P2 emission originates from the charge-transfer state (Fig. 4g)[40,41].

It is worth noting that the transformation from HO-PBA dimer to trimer occurs rapidly at room temperature. Following addition of water to the THF−$d_8$ solution of HO-PBA dimer, a $^1$H NMR measurement was performed immediately. The resulting $^1$H NMR spectrum clearly indicated that HO-PBA dimer had completely transformed into trimer at room temperature, and the results of $^1$H NMR, ESI-Q-TOF MS and UV–Raman measurements suggest HO-PBA trimers remain stable under ambient environments during the 7-day study (Supplementary Fig. 12). The same changes were also observed for CH$_3$-HO-PBA and CF$_3$-HO-PBA dimers (Supplementary Figs. 12–18). Moreover, the results of fluorescence, NMR and UV–Raman measurements indicate that the boroxine structure remains stable over a wide pH range, specifically, 2.4 <pH <9 for HO-PBA trimer (Fig. 4h and Supplementary Fig. 19a, b), and 2.4 <pH <10 for CH$_3$-HO-PBA trimer (Supplementary Figs. 13h and 19c, d). The remarkable stability of the boroxine structure in water and its wide pH tolerance range make it suitable for various applications where water is present.

### Role of water in the formation of the boroxine structure

The transformation of HO-PBA dimer to trimer in water prompts an interesting question: does water participate in this process? To address this question, the isotope tracing method was employed by dissolving HO-PBA dimer in a methanol−H$_2$$^{18}$O (5:1, v/v) solution and subsequently measuring ESI-Q-TOF MS. In the methanol−H$_2$O solution, HO-PBA trimer exhibits two peaks at $m/z$ 359.13 and 373.14 (Fig. 4a), whereas in the methanol−H$_2$$^{18}$O solution, the peaks shift to $m/z$ 365.12 and 377.13 (Fig. 4i). The $m/z$ 365.12 corresponds to the molecular weight of the deprotonated HO-PBA trimer with three $^{18}$O ([T\3$^{18}$O − H]$^-$), while the $m/z$ 377.13 corresponds to the molecular weight of the deprotonated product with two $^{18}$O ([T\2$^{18}$O + CH$_2$ − H]$^-$), resulting from the etherification of HO-PBA trimer and the solvent methanol during the MS measurement. This result implied that H$_2$$^{18}$O participated in the transformation process of HO-PBA dimer to trimer.

The involvement of H$_2$O in the transformation process of HO-PBA dimer to trimer, along with the shared empirical formula between the two species, suggests a potential catalytic function of H$_2$O in this process. To validate this hypothesis, we conducted a NMR titration experiment by gradually introducing D$_2$O into a solution of HO-PBA dimer in THF−$d_8$. With increasing amounts of D$_2$O, the NMR signals corresponding to the aromatic protons of HO-PBA dimer (marked with red circles) gradually diminish. Simultaneously, signals corresponding to the aromatic protons of HO-PBA trimer (marked with blue triangles) emerge and gradually intensify (Fig. 5a). Eventually, the signals of HO-PBA dimer disappear, leaving only the signals of HO-PBA trimer in the $^1$H NMR spectra. This indicates the complete conversion of HO-PBA dimers into trimers. Under ambient environments, upon removal of H$_2$O, the reaction spontaneously proceeds in the opposite direction, resulting in the regeneration of HO-PBA dimers (Supplementary Fig. 20). These results demonstrate that D$_2$O (or H$_2$O) acts as a reactant rather than a catalyst in the transformation process from HO-PBA dimer to trimer (Supplementary Fig. 21), and HO-PBA dimers undergo a reversible reaction with D$_2$O (or H$_2$O), producing HO-PBA trimer–H$_2$O complexes (simplified as HO-PBA trimer for convenience).

The transformation from HO-PBA dimer to trimer was further investigated using $^2$D NMR spectroscopy. In the $^2$D NMR spectrum of THF, no peaks are observed due to the low deuterium content of THF. After adding a trace amount of D$_2$O (2 μL) into THF (500 μL), a distinct peak attributed to D$_2$O appears at 2.4 ppm. With increasing amounts of D$_2$O, this peak progressively intensifies and shifts towards lower field (Supplementary Fig. 22). Interestingly, after adding a trace amount of D$_2$O (2 μL) into a THF solution of HO-PBA dimer (0.27 mol L$^{-1}$, 500 μL), no peak at 2.4 ppm emerges. Instead, three peaks labeled as n, m, and x appear in the range of 7–9.5 ppm (Fig. 5b), indicating the presence of three types of active hydrogen in the solution. The peak "m" is assigned to the deuterium of the boron hydroxyl group in HO-PBA dimer, which gradually disappears with increasing amounts of D$_2$O due to the transformation of HO-PBA dimer to trimer. The peak "n" is attributed to the deuterium of phenolic hydroxyl groups in both HO-PBA dimer and trimer, and they could not be discriminated due to their similar chemical shifts. The peak "x" should be assigned to D$_2$O in HO-PBA trimer–D$_2$O complex, with its chemical shift changing from 2.4 to 7.5 ppm as a result of the binding interaction with HO-PBA trimer. Furthermore, upon addition of more D$_2$O, the binding sites of D$_2$O in HO-PBA trimer become saturated. Consequently, a signal corresponding to free D$_2$O emerges at 3.3 ppm.

Subsequently, to investigate the binding site of D$_2$O, $^{11}$B NMR spectroscopy was adopted, which is sensitive to the coordination environment of boron atoms[42]. If the binding site of D$_2$O was on the boron atom, the chemical shift of HO-PBA trimer would move to higher field compared to that of HO-PBA dimer. However, it exhibits a light shift towards lower field (Fig. 5c), indicating that the binding site of D$_2$O is not on the boron atom. Therefore, D$_2$O might bind to HO-PBA trimer through hydrogen bond formed between the deuterium atom of D$_2$O and oxygen atom of the boroxine structure in HO-PBA trimer (Fig. 5d).

In the $^2$D NMR spectrum of PBA and phenol in the THF−D$_2$O solution, no signals corresponding to the bound D$_2$O are observed (Supplementary Fig. 23), indicating the non-formation of PBA–D$_2$O and phenol–D$_2$O complex. Therefore, the capability to form a complex by binding with D$_2$O is a distinctive characteristic of the boroxine structure present in HO-PBA trimer.

Last, the number of D$_2$O bound to a molecule of HO-PBA trimer was determined. In the case of $n = 2$ (Supplementary Fig. 21), the equilibrium constants, calculated by the integrating the peak areas of HO-PBA dimer at 7.54 ppm and HO-PBA trimer–D$_2$O complex at 7.64 ppm (Fig. 5a), remain constant as the amount of D$_2$O increases (Supplementary Table 11). Based on this, it can be inferred that a HO-PBA trimer binds to two molecules of D$_2$O (Fig. 5d). Furthermore, through a variable-temperature $^1$H NMR study ranging from −20 °C to 25 °C (Supplementary Fig. 24a), the thermodynamic parameters of the transformation from HO-PBA dimer to trimer were determined as follows: $\Delta H = -20.10$ kJ mol$^{-1}$ and $\Delta S = -6.46$ J mol$^{-1}$ K$^{-1}$ (Supplementary Fig. 24b), indicating that the transformation from HO-PBA dimer to trimer is enthalpically driven.

### Water-compatible DCBs

DCBs have been firmly integrated into diverse research fields, such as organic synthesis, material science, and biomedicine. Nonetheless, most of DCBs are water-incompatible, which restricts their applicability to certain specific fields[33,43]. Herein, we reveal that B−O bonds in HO-PBA trimer are water-compatible DCBs, providing a significant member to the DCBs family.

The dynamic nature of B−O bonds in the water-stable boroxine structure was illustrated by the rapid exchanges between various HO-PBA trimeric structures. To carry out the exchange reaction, HO-PBA trimers and CF$_3$-HO-PBA trimers were co-dissolved in a methanol−water solution at room temperature, and the solution was then analyzed by ESI-Q-TOF MS. In addition to the previously identified peaks of HO-PBA trimer ([T − H]$^-$ at $m/z$ 359.13 and [T + CH$_2$ − H]$^-$ at $m/z$ 373.14) and CF$_3$-HO-PBA trimer ([T3 − H]$^-$ at $m/z$ 563.11 and

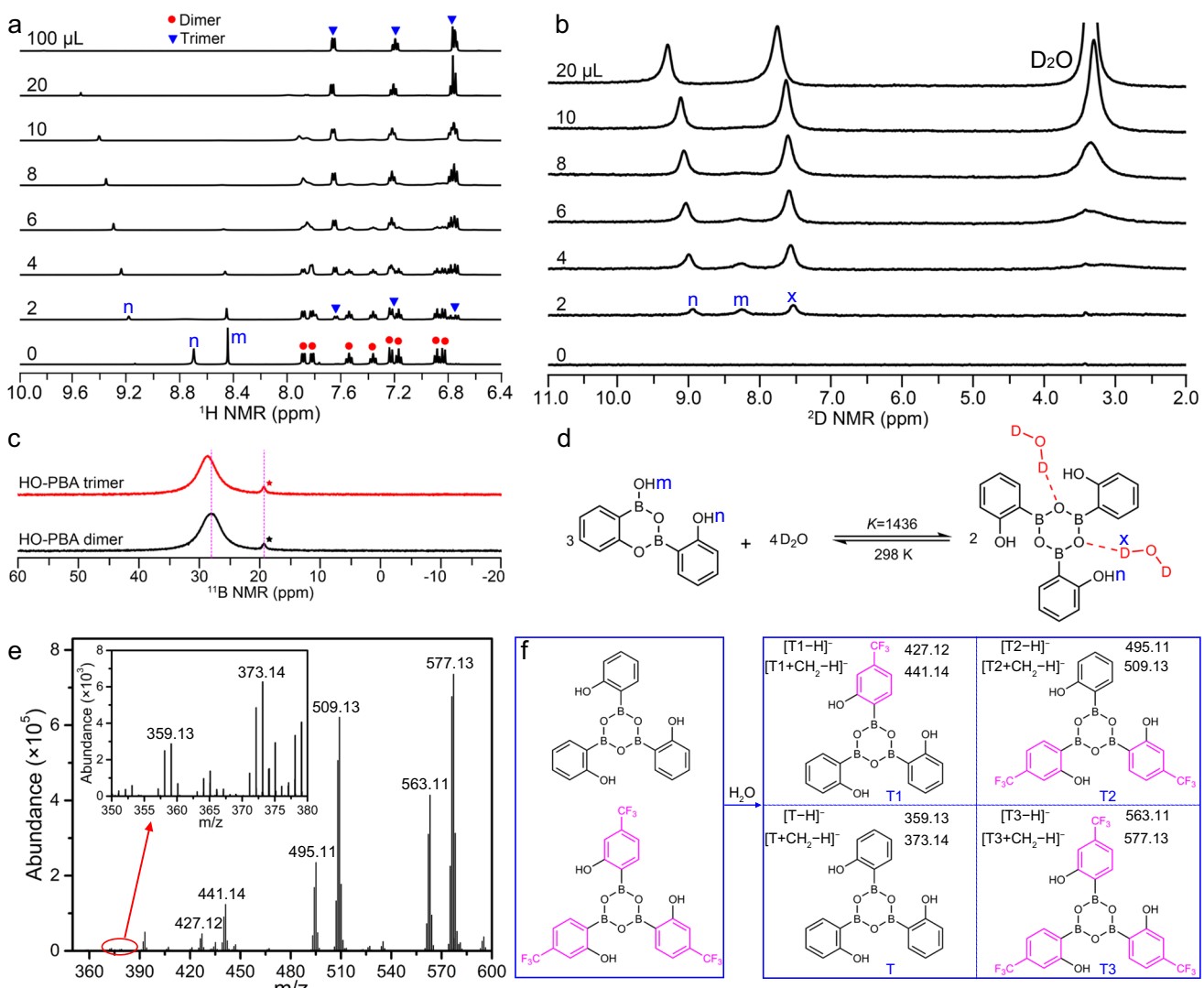

**Fig. 5 | Role of water in the formation of the boroxine structure and water-compatible DCBs. a, b** $^1$H (**a**) and $^2$D (**b**) NMR spectra of equilibrium mixture of HO-PBA dimer and trimer with various amount of $D_2O$. The initial sample was prepared by dissolving 32.8 mg HO-PBA dimer in 500 μL THF−$d_8$ (**a**) or THF (**b**). **c** $^{11}$B NMR spectra of HO-PBA dimer (black line) in THF−$d_8$ and HO-PBA trimer (red line) in $D_2O$−THF−$d_8$ (1:5, v/v) mixture. The chemical shifts of $^{11}$B were determined relative to 0.1 mol L$^{-1}$ boric acid (19.4 ppm, marked with a star)[51], which served as the

external reference. **d** Reversible reaction between HO-PBA dimer and $D_2O$ to produce HO-PBA trimer−$D_2O$ complex. **e** ESI-Q-TOF mass spectrum of products from mixing HO-PBA dimers and $CF_3$-HO-PBA dimers together in a methanol−water (5:1, v/v) solution, acquired in a negative mode. **f** Illustration of the dynamic exchange between HO-PBA trimer (T) and $CF_3$-HO-PBA trimer (T3). The exchange products are denoted by T1 and T2.

[T3 + CH$_2$ − H]$^-$ at *m/z* 577.13), peaks related to their exchange products ([T1 − H]$^-$ at *m/z* 427.12, [T1 + CH$_2$ − H]$^-$ at *m/z* 441.14, [T2 − H]$^-$ at *m/z* 495.11, and [T2 + CH$_2$ − H]$^-$ at *m/z* 509.13) were also present in the mass spectrum (Fig. 5e,f). This suggested an exchange reaction occurred between HO-PBA trimer and $CF_3$-HO-PBA trimer, resulting in a mixture of homo- and hetero-substituted HO-PBA trimeric structure under equilibrium. Similar exchange reactions were also observed between HO-PBA trimer and $CH_3$-HO-PBA trimer (Supplementary Fig. 25), as well as between $CF_3$-HO-PBA trimer and $CH_3$-HO-PBA trimer (Supplementary Fig. 26). These results verified the highly dynamic nature of B−O bonds in the boroxine structure in aqueous conditions at room temperature. The water-incompatible B−O bonds in the conventional boroxine structure are also dynamic, however, to carry out the exchange reactions, their toluene solutions need to be heated at 60 °C for 8 h[44]. By comparison, the water-compatible B−O bonds in the water-stable boroxine structure can rapidly exchange at room temperature, which suggests the flexibility of these structures in various applications.

## Recognition of F$^-$ by the boroxine structure

Boroxine rings have been identified as effective anion receptors for F$^-$ in lithium ion batteries owing to their unique ring structures and electron-deficient properties[4,45]. However, the applicability of boroxine rings as anion receptors in aqueous media is limited due to their hydrolytic instability[45]. Fortunately, the discovery of the water-stable boroxine structure provides a feasible solution to this challenge.

As shown in Fig. 6a, the $^1$H NMR spectrum of HO-PBA trimer contains four sets of peaks, whereas more than ten sets of peaks appear after the addition of F$^-$. In contrast, the addition of chloride ions (Cl$^-$), bromide ions (Br$^-$), or iodine ions (I$^-$) did not cause any alterations to the $^1$H NMR spectrum of HO-PBA trimer. These results confirmed that HO-PBA trimer could selectively recognize F$^-$ in aqueous media.

The binding affinity of HO-PBA trimer to F$^-$ in aqueous media was evaluated by comparing it with the commonly used F$^-$ recognition molecule, PBA[25-27]. Initially, a NaF suspension was prepared in a $H_2O$−dimethylsulfoxide (DMSO) (1:6, v/v) mixture (Fig. 6b, left panel).

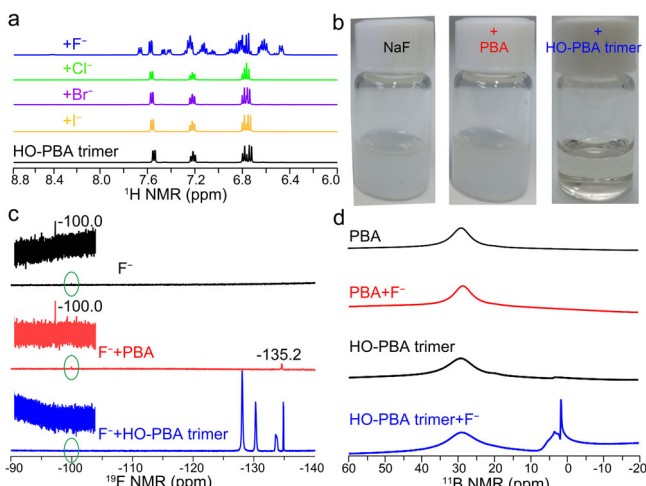

**Fig. 6 | Ultra-strong binding capacity of the boroxine structure toward F⁻ in aqueous media.** **a** ¹H NMR spectra of HO-PBA trimer (120 mmol L⁻¹) before (black) and after the addition of one molar equivalent of NaF (blue), NaCl (green), NaBr (purple), or NaI (yellow). **b**, **c** Photos (**b**) and ¹⁹F NMR spectra (**c**) of NaF (120 mmol L⁻¹) before (black) and after addition of one molar equivalent of PBA (red), or HO-PBA trimer (blue). **d** ¹¹B NMR spectra of PBA (120 mmol L⁻¹) or HO-PBA trimer (120 mmol L⁻¹) before and after addition of one molar equivalent NaF. All NMR measurements were carried out at room temperature using D₂O–DMSO–$d_6$ (1:6, v/v) mixtures as solvents.

In the ¹⁹F NMR spectrum of the NaF suspension (Fig. 6c, black line), a weak signal at −100.0 ppm attributed to free F⁻ is observed, indicating that only a small amount of F⁻ has dissolved in the suspension. Upon adding one molar equivalent of PBA, the NaF/PBA remained as a suspension (Fig. 6b, middle panel). In the ¹⁹F NMR spectrum of the NaF/PBA suspension (Fig. 6c, red line), a new yet weak signal at −135.2 ppm attributed to the PBA–F⁻ complex appears, indicating that only a small amount of PBA–F⁻ complex was formed in the NaF/PBA suspension. Due to the low content of the PBA–F⁻ complex, the ¹¹B NMR spectrum of the NaF/PBA suspension (Fig. 6d, red line) shows no change compared with that of the PBA solution (Fig. 6d, black line, Top). In sharp contrast, the NaF suspension became clear after adding one molar equivalent of HO-PBA trimer (Fig. 6b, right panel). In the ¹⁹F NMR spectrum of the NaF/HO-PBA trimer solution (Fig. 6c, blue line), the signal of free F⁻ disappears, and multiple strong signals resulting from the equilibrations of the HO-PBA trimer–F⁻ complex and HO-PBA trimer (Supplementary Fig. 27)[46] appear in the −125 to −136 ppm range. Correspondingly, in the ¹¹B NMR spectrum of the NaF/HO-PBA trimer solution (Fig. 6d, blue line), strong sp³–¹¹B signals of HO-PBA trimer–F⁻ complex (3.1 and 1.2 ppm) are observed, apart from the sp²–¹¹B signal (28.7 ppm)[42,46]. Similar results were obtained for CH₃-HO-PBA trimer (Supplementary Fig. 28) and CF₃-HO-PBA trimer (Supplementary Fig. 29). These findings indicate that the boroxine structure exhibits significantly stronger binding affinity to F⁻ than PBA does, illustrating its good potential in F⁻ detection and separation.

## Boroxine-based dynamic hydrogel

Hydrogels have gained significant attention in recent years because of their potential applications in tissue engineering, drug delivery, biosensing, and other areas[47,48]. Herein, we designed a boroxine-based hydrogel to demonstrate the exceptional performance of the boroxine structure as cross-linkers. First, a hydrophilic copolymer poly(-poly(ethylene glycol) methyl ether acrylate₀.₂₈ₘ-*co*-glycidyl methacrylateₘ), abbreviated to poly(PEGMEA₀.₂₈ₘ-*co*-GMAₘ), was synthesized using reversible addition-fragmentation chain transfer polymerization technique (Supplementary Fig. 30). The number average molecular weight (Mₙ) and polydispersity index (PDI) of the

copolymer were determined to be 29, 800 g mol⁻¹ and 1.58, respectively, by gel permeation chromatography (Supplementary Fig. 31). Subsequently, ethylenediamine was decorated to poly(PEGMEA₀.₂₈ₘ-*co*-GMAₘ) through a ring-opening reaction with the epoxide moiety in GMA, which yielded poly(PEGMEA₀.₂₈ₘ-*co*-AMAₘ) (Supplementary Fig. 30). Finally, poly(PEGMEA₀.₂₈ₘ-*co*-AMAₘ) was functionalized with PBA and 2-methoxy phenylboronic acid (CH₃O-PBA), respectively, through the Schiff-base reactions between amino group and aldehyde group, generating two corresponding copolymers, namely, poly(PEGMEA₀.₂₈ₘ-*co*-AMAₘ-PBA₀.₅₇ₘ) and poly(PEGMEA₀.₂₈ₘ-*co*-AMAₘ-(CH₃O-PBA)₀.₃₉ₘ) (Supplementary Fig. 30). A third copolymer, poly(PEGMEA₀.₂₈ₘ-*co*-AMAₘ-(HO-PBA)₀.₃₉ₘ), was obtained by adopting BBr₃ to convert the methoxy group of CH₃O-PBA in poly(PEGMEA₀.₂₈ₘ-*co*-AMAₘ-(CH₃O-PBA)₀.₃₉ₘ) into a hydroxyl group (Supplementary Fig. 30). The detailed characterization data of these polymers are described in Supplementary Fig. 32.

Poly(PEGMEA-*co*-AMA-PBA) and poly(PEGMEA-*co*-AMA-(CH₃O-PBA)) are both water soluble due to the hydrophilic nature of poly(-PEGMEA-*co*-AMA) and the favorable solubility of PBA and CH₃O-PBA in water (Fig. 7b). Interestingly, after the methoxy group in CH₃O-PBA of poly(PEGMEA-*co*-AMA-(CH₃O-PBA)) was converted to the hydroxyl group, the resulted poly(PEGMEA-*co*-AMA-(HO-PBA)) formed a hydrogel when mixed with water, as shown in Fig. 7b.

The three copolymer/water mixtures were further analyzed using low-field nuclear magnetic resonance (LF–NMR) to measure the transverse spin-spin relaxation time (T₂) of water in each mixture. The T₂ distribution curve of each mixture displays three water components (Fig. 7c), indicating the presence of three forms of water with different degrees of freedom. The first component, T₂₁ with the shortest relaxation time, corresponds to the protons of bound waters that are combined with the polymer molecules. The second component, T₂₂, corresponds to protons of immobilized waters that are confined in the space between the polymer chains. The third component, T₂₃ with the longest relaxation time, corresponds to protons of free waters[49]. In the poly(PEGMEA-*co*-AMA-PBA)/water and poly(PEGMEA-*co*-AMA-(CH₃O-PBA))/water mixture, the T₂₃ peak dominates the LF–NMR spectra with the proportionate peak areas of 80.3% and 65.5%, respectively (Supplementary Fig. 33). This indicates that water in these mixtures primarily exists as free water. In contrast, the T₂₂ peak with proportionate peak area of 88.6% dominates the LF–NMR spectrum of poly(PEGMEA-*co*-AMA-(HO-PBA))/water mixture (Supplementary Fig. 33), which agrees with the well-established knowledge that water in the hydrogel is mainly confined within its 3D polymeric network.

The formation of poly(PEGMEA-*co*-AMA-(HO-PBA)) hydrogel was believed not to be driven by the noncovalent interactions within the copolymers, since hydrogels did not form by its counterparts poly(-PEGMEA-*co*-AMA-PBA) and poly(PEGMEA-*co*-AMA-(CH₃O-PBA)). Instead, the boroxine structures, which are derived from HO-PBA and crosslink the poly(PEGMEA-*co*-AMA-(HO-PBA)) chains, are responsible for the hydrogel formation. This presumption is supported by the fact that the hydrogel remains stable in both acidic (pH=2) and alkaline (pH=10) solution, as shown in Fig. 7b. This stability is consistent with the finding that the boroxine structure remains stable over a wide range of pH values (Fig. 4h). Moreover, the hydrogel could undergo a reversible gel–sol transition due to the dynamic nature of the boroxine structures. As shown in Fig. 7d, the hydrogel collapses in the presence of excess HO-PBA trimer (photo marked with a triangle), which results from the de-crosslinking of the polymer networks via an exchange reaction of free HO-PBA trimers with the boroxine cross-linkers (Fig. 7e). Removal of HO-PBA trimers through washing with excess THF resulted in the reconstruction of the networks through the boroxine structures, thereby reforming the hydrogel (Fig. 7d, photo marked with a circle, Fig. 7e). Notably, the hydrogel was not influenced by the addition of excess PBA (Fig. 7d, photo marked with a square), which highlighted the role of HO-PBA. Overall, the poly(PEGMEA-*co*-AMA-

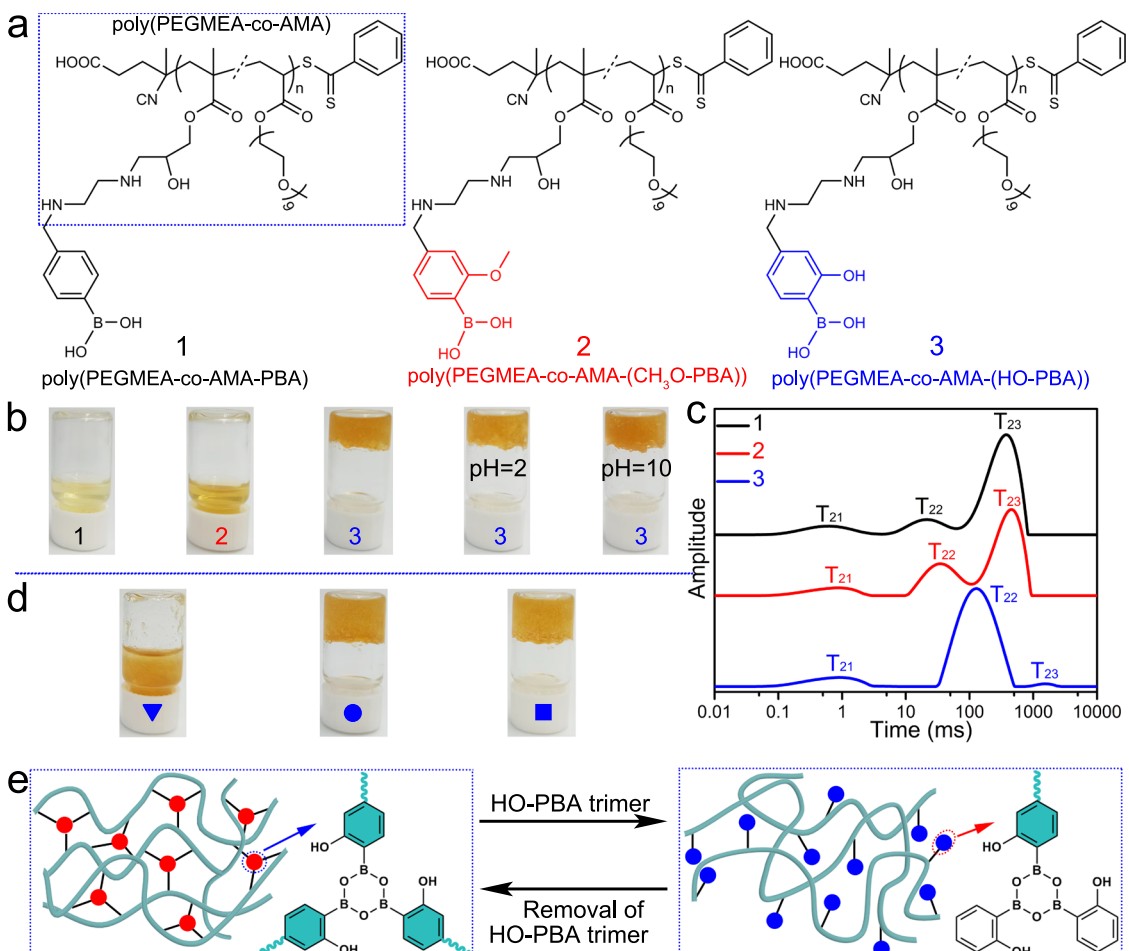

**Fig. 7 | A boroxine-based hydrogel with high acid–base stability and reversible gel–sol transition. a** Molecular structures of poly(PEGMEA-*co*-AMA-PBA) (**1**), poly(PEGMEA-*co*-AMA-(CH₃O-PBA)) (**2**), and poly(PEGMEA-*co*-AMA-(HO-PBA)) (**3**). **b** Photos of **1**, **2**, and **3** in water, as well as **3** in acidic (pH=2) or alkaline (pH=10) solution, concentrations: 50 mg mL⁻¹. **c** $T_2$ distribution curves of **1** (black), **2** (red), and **3** (blue) in water at room temperature, recorded by LF–NMR, concentrations: 50 mg mL⁻¹. **d** Photos of **3** in the presence (triangle) and absence (circle) of HO-PBA trimer, as well as **3** in the presence of PBA (square) in a water–THF (v/v: 2:1) mixture. The molar ratios of PBA and HO-PBA trimer to HO-PBA fragments in **3** are approximately 3:1 and 1:1, respectively. **e** Illustration of the reversible gel–sol transition of **3** regulated by HO-PBA trimer.

(HO-PBA)) hydrogel represents a boroxine-based hydrogel with high acid–base stability and reversible gel–sol transition, providing a design strategy for hydrogel materials.

## Discussion

In summary, we discover a water-stable boroxine structure, with excellent pH stability and water-compatible DCBs capable of undergoing exchanges at room temperature, fundamentally addressing the long-standing challenge of hydrolytic instability of boroxines. With this discovery, we achieves the selective recognition of F⁻ by boroxines in aqueous media, with a significantly stronger affinity than the widely used F⁻ receptor PBA, which provides a avenue to develop advanced materials and techniques for F⁻ detection and separation. Moreover, we develop a hydrogel cross-linked by the water-stable boroxines, which exhibits high acid–base stability and reversible gel–sol transition endowed by the excellent pH stability and dynamic nature of the boroxine structures. This pioneering achievement introduces a strategy for designing functionally dynamic hydrogels. This work is just the beginning, as the water-stable boroxine structure holds great potential for many other fields beyond the above preliminary applications, including biosensing, bioseparation, drug design, hydrophilic COFs and molecular architectures, as well as repairable underwater adhesive. We believe that future researches in various fields will continue to produce remarkable results based on the water-stable boroxine structure with DCBs.

## Methods

### Materials

Ethylenediamine (>99%), 2-hydroxyphenylboronic acid (HO-PBA, 97%), 10-hydroxybenzo[*h*]quinoline (HBQ, 98%), 4-formylphenylboronic acid (97%), boron tribromide (99.9% metals basis), and glycidyl methacrylate (GMA, 97%) were purchased from Aladdin reagent (Shanghai, China). 2-hydroxy-5-methylphenylboronic acid (CH₃-HO-PBA, 97%) was purchased from Bide Pharmatech (Shanghai, China). 2-hydroxy-4-(trifluoromethyl)phenylboronic acid (CF₃-HO-PBA, 98%) and (4-Formyl-2-methoxyphenyl)boronic acid (98%) were purchased from Leyan (Shanghai, China). 4-cyano-4-(phenylcarbonothioylthio)pentanoic acid (CPA, >97%), dry methanol (99.9%, Water ≤50 ppm) and dry acetonitrile (99.9%, Water≤50 ppm) were purchased from macklin (Shanghai, China). Dry dichloromethane (99.9%, Water ≤30 ppm) was obtained from J&K Scientific. Poly(ethylene glycol) methyl ether acrylate (PEGMEA, average $M_n$ 480) was purchased from Sigma-Aldrich. Azobisisobutyronitrile (AIBN) was obtained from Tianjin guangfu fine chemical research institute. Other reagents and solvents were obtained from commercial suppliers. Unless otherwise noted, chemicals were used as received.

## Single crystal growth

The single crystals of HO-PBA, CF$_3$-HO-PBA and CH$_3$-HO-PBA were grown through slow evaporation of the water–ACN (1:1, v/v) solution of HO-PBA, the water–acetone (2:1, v/v) of CF$_3$-HO-PBA and the water–ACN (2:1, v/v) solution of CH$_3$-HO-PBA, respectively.

## Synthesis of poly(PEGMEA-*co*-GMA)

PEGMEA (9.60 g, 20 mmol), GMA (5.68 g, 40 mmol), CPA (140 mg, 0.5 mmol) and AIBN (10.5 mg, 0.064 mmol) were dissolved in 100 mL of N,N-dimethylformamide (DMF), and the solution was stirred at 65 °C for 48 h in an argon atmosphere after three freeze–pump–thaw cycles. The crude product was purified via dialysis in deionized water (molecular weight cut-off, 3500). Poly(PEGMEA-*co*-GMA) was obtained by vacuum freeze-drying.

## Synthesis of poly(PEGMEA-*co*-AMA)

Poly(PEGMEA-*co*-GMA) (6 g) and ethylenediamine (40 mL) were dissolved in 25 mL of DMF, and the solution was stirred at 65 °C for 10 h in a nitrogen atmosphere. After being concentrated by using the vacuum rotary evaporation, the crude product was purified via dialysis in deionized water (molecular weight cut-off, 3500). Poly(PEGMEA-*co*-AMA) was obtained by vacuum freeze-drying.

## Synthesis of poly(PEGMEA-*co*-AMA-(CH$_3$O-PBA)) and poly(-PEGMEA-*co*-AMA-PBA)

Poly(PEGMEA-*co*-AMA) (2 g) and (4-Formyl-2-methoxyphenyl)boronic acid (1 g, 5.6 mmol) were mixed in 35 mL of dry methanol, and the reaction was carried out at 60 °C under stirring for 25 h in a nitrogen atmosphere. Subsequently, the reaction mixture was cooled, and sodium borohydride (2 g, 52.6 mmol) was added in small portions in an ice bath. After addition of sodium borohydride, the reaction mixture was stirred for 15 h at room temperature, and then was quenched by adding 10 mL of methanol. After being concentrated by using the vacuum rotary evaporation, the crude product was purified via dialysis in deionized water (molecular weight cut-off, 3500). Poly(PEGMEA-*co*-AMA-(CH$_3$O-PBA)) was obtained by vacuum freeze-drying. The synthesis process of poly(PEGMEA-*co*-AMA-PBA) was similar to that of poly(PEGMEA-*co*-AMA-(CH$_3$O-PBA)).

## Synthesis of poly(PEGMEA-*co*-AMA-(HO-PBA))

Poly(PEGMEA-*co*-AMA-(CH$_3$O-PBA)) (1.2 g) was dispersed in dry dichloromethane (DCM, 75 mL) in a nitrogen atmosphere, and the reaction mixture was stirred at −78 °C for 1 h after the addition of BBr$_3$ (1.0 M in DCM, 12 mL) by using a constant pressure drip funnel. Then, the reaction mixture was stirred for 3 h at room temperature, followed by the addition of 3 mL of cold water to quench the reaction. The generated precipitation was collected, and purified via dialysis in deionized water (molecular weight cut-off, 3500). Poly(PEGMEA-*co*-AMA-(HO-PBA)) was obtained by vacuum freeze-drying.

## Characterization

Fluorescence spectra were measured on a PerkinElmer FL6500 fluorescence spectrometer. The fluorescence spectra of liquid samples were measured at right angles to the incident radiation, which was incident on samples in a 1 cm path–length quartz cell. For solid samples, the fluorescence spectra were measured using a solid sample holder positioned diagonally at −45° with respect to the incident and output beams. In the HBQ experiment, fluorescence samples, solutions of HBQ and a mixture of HBQ and HO-PBA dimer, were prepared at room temperature, with concentrations of $4.0 \times 10^{-5}$ mol L$^{-1}$ for HBQ and $2.0 \times 10^{-4}$ mol L$^{-1}$ for HO-PBA dimer in the two solutions. Water–THF mixtures in a ratio of 4:1 (v/v) were adopted as solvents. All the Fluorescence spectra were recorded at room temperature with both excitation and emission slits set to 5 nm. Fluorescence absolute quantum yields of the liquid and solid samples were determined by an Edinburgh FLS980 spectrometer with an integrating sphere. SEM measurements were carried out on a JEOL JSM − 7800F instrument. For the liquid samples, SEM samples were prepared by dropping one droplet of water–THF (2:3, v/v) solution of HO-PBA, CH$_3$-HO-PBA or CF$_3$-HO-PBA onto clean silicon wafers and evaporating the solvent at room temperature and in open air. For solid samples, SEM samples were prepared by directly sticking the powder on the carbon tapes. XRD patterns were collected by a PANAlytical X'pert Pro Powder X−ray diffractometer. The single−crystal XRD data were collected on a Bruker D8 Venture CMOS−based diffractometer (Mo–Kα radiation, $\lambda = 0.71073$ Å) using the SMART and SAINT programs at 120 K. Final unit cell parameters were based on all observed reflections from integration of all frame data. The structures were solved in the space group by direct method and refined by the full−matrix least−squares using SHELXTL − 97 fitting on F$^2$. All non−hydrogen atoms were refined anisotropically. The hydrogen atoms were located geometrically and assigned fixed isotropic thermal parameters[50]. Variable-temperature NMR spectra were recorded on a JEOL JNM-ECZL400S spectrometer, and other NMR experiments were performed on a Bruker AVANCE III 400 spectrometer. To investigate the selective association of the water-stable boroxine structure with F$^-$, solutions of HO-PBA trimer, NaF suspension, and a mixture of HO-PBA trimer and NaF were prepared for $^1$H, $^{11}$B, and $^{19}$F NMR measurements. The concentrations of HO-PBA trimer and NaF were both 120 mmol·L$^{-1}$, and D$_2$O–DMSO−$d_6$ mixtures in a ratio of 1:6 (v/v) were adopted as solvents. For CH$_3$-HO-PBA trimer and CF$_3$-HO-PBA trimer, similar NMR experiments were carried out. An AB SCIEX TOF/TOF 5800 equipped with a neodymium: yttrium aluminum garnet (Nd: YAG) laser (349 nm wavelength) was used for MALDI-TOF MS experiments. Trans-2-[3-(4-tert-butylphenyl)-2-methyl-2-propenylidene]malononitrile was used as a MALDI matrix. Silver trifluoroacetate was used as cationization reagent. To investigate the exchange reaction between different dimers, the sample was prepared by co-dissolving any two of HO-PBA dimer, CH$_3$-HO-PBA dimer, and CF$_3$-HO-PBA dimer (each at a concentration of 20 mg mL$^{-1}$) in dry THF at room temperature. The matrix (20 mg mL$^{-1}$) and cationization reagent (10 mg mL$^{-1}$) were, respectively, dissolved in dry THF. These solutions were mixed at a volume ratio of sample: matrix: cationization agent =1:10:1. The mixture solution (0.5 μL) was deposited on a MALDI sample plate, and the spots were dried in air at room temperature. The spectra were obtained in a negative reflector mode. The ESI-Q-TOF mass spectra were obtained on an Agilent 6540 Q-TOF mass spectrometer, and acetonitrile was used as mobile phase. Samples were dissolved in the solvent and ionized using an ESI source in a negative mode. The ESI source parameters were as following: gas temperature 325 °C, drying gas 8 L min$^{-1}$, nebulizer 35 psig, capillary voltage 3500 V and m/z range 20–2000. To examine the conversion from dimers to trimers, HO-PBA dimers (1.7 mg mL$^{-1}$) were dissolved in a methanol–water (5:1, v/v) mixture at room temperature. To investigate the exchange reaction between different trimers, any two of HO-PBA trimer, CH$_3$-HO-PBA trimer, and CF$_3$-HO-PBA trimer (each at a concentration of 1.7 mg mL$^{-1}$) were co-dissolved in a methanol–water (5:1, v/v) mixture at room temperature. In the isotope tracing experiment, HO-PBA dimers (1.7 mg mL$^{-1}$) were dissolved in a methanol–H$_2^{18}$O (5:1, v/v) mixture at room temperature. All samples were filtered using 0.22 μm pore-size membrane filters. UV–Raman spectra were recorded using a home-made single stage UV–Raman spectrograph with a spectral resolution of 2 cm$^{-1}$. The single-frequency UV laser line at 244 nm was from an efficient external cavity frequency doubler (Wavetrain, Spectra-Physics) of the single-frequency laser at 488 nm laser (Genesis, CX 488, Coherent). The exciting source was with an output of 10 mW and the power of the laser at the sample was about 5.0 mW. Gel permeation chromatography (GPC) was performed on a PL-GPC220 chromatograph with N,N-dimethylformamide as the solvent. Fourier transform infrared (FT-IR) spectra were recorded on a Nicolet iS50 FT-IR spectrometer equipped with an attenuated total

reflection accessory. A MesoMR23-060V-I NMR analyzer (Suzhou Niumag Analytical Instrument Corporation, Suzhou, China) with a permanent magnet having resonance frequency 21 MHZ was used to measure $T_2$. The boron contents in poly(PEGMEA-*co*-AMA-PBA) and poly(PEGMEA-*co*-AMA-(HO-PBA)) were measured by using a Perki-nElmer optima 8000 inductively coupled plasma optical emission spectrometer. The free energy calculation was conducted by DFT method with M06-2X functional and TZVP basis set using the Gaussian 16 program (Revision A.03). The calculation of fluorescent spectrum of HO-PBA trimer was performed by DFT method with B3LYP density functional and 6−31 G* basis set using the Gaussian 16 program (Revision A.03).

## Data availability

The data supporting the main findings of this work are available within the paper and its Supplementary Information, or available from the corresponding author upon request. Crystallographic data for the structures reported in this article have been deposited at the Cambridge Crystallographic Data Centre, under the deposition numbers CCDC 2280484, 2280485 and 2280486. These data can be obtained free of charge via https://www.ccdc.cam.ac.uk/structures/. Source data are provided with this paper.

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

## Acknowledgements

This work was supported by the National Key R&D Program of China (Grant No. 2022YFC3400800, G. Qing), the National Natural Science Foundation of China (21922411 and 22174138, G. Qing), DICP Innovation Funding (DICP-I202243 and I202229, G. Qing), the Dalian Outstanding Young Scientific Talent (2020RJ01, G. Qing), and grant from the Department of Education of Liaoning (General Program, Grant No. LJKMZ20220902, Y. Zhang). We thank associate Prof. Jingfeng Han (Dalian Institute of Chemical Physics) for assistance with UV–Raman spectroscopy, associate Prof. Xiuying Wang (Dalian Polytechnic University) for assistance with NMR measurements, and Prof. Mingqian Tan (Dalian Polytechnic University) for assistance with LF–NMR measurements.

## Author contributions

X. Li and Y. Zhang (Yongjie Zhang) contributed equally to this work. X. Li, Y. Zhang (Yongjie Zhang) and G. Qing conceived the study, designed the experiments and wrote the manuscript. X. Li and Y. Zhang (Yongjie Zhang) performed most of the experiments. Z. Shi designed the synthesis of polymers and helped in polishing the language. D. Wang designed the theoretical calculations. H. Yang helped in the exchange reaction experiments. Y. Zhang (Yahui Zhang) and H. Qin helped in the NMR measurement. W. Lu, J. Chen and Y. Li helped in the synthesis of polymers.

## Competing interests

The authors declare no competing interests.
