## [Peer Review File · Nature Communications]

REVIEWER COMMENTS

Reviewer #1 (Remarks to the Author):

The chemical instability of boroxines to hydrolysis is an intrinsic problem. The manuscript presented by Qing and coworkers reports an unprecedented water-stable boroxine. They found that HO-PBA exists as a dimer and interestingly the dimer transformed into a trimer (boroxine) upon exposure to water. The structure of trimer was carefully characterized and discussed by several techniques such as NMR, MS, UV-Raman, fluorescence, reactions with HBQ, Br₂ or Fe³⁺, DFT calculations, although X-ray structure of boroxine was not clarified. Furthermore, the trimer (boroxine) was found to show excellent stability against water in the pH range from 2 to 9 and unique dynamic behavior with rapid exchange between various trimers. Taking advantage of the stability and dynamic behavior, the boroxine-based dynamic hydrogel was demonstrated.

These findings are very significant because they change the common knowledge about the hydrolytic instability of boroxines and represent a major breakthrough not only in the fundamental properties of boroxines but also in boroxine materials. Although the role of water in the conversion from dimer to trimer (boroxine) and the reason for the high chemical stability of trimer (boroxine) with water are not clear, the reviewer hopes these questions will be elucidated soon. The significance and novelty of this work are valuable for the scientific community. For all these reasons, I recommend the publication of this work after some minor revisions.

1. Is it possible to observe a monomer (boronic acid) of HO-PBA by ¹H NMR? Can dimer or trimer be hydrolyzed completely to monomer (HO-PBA)? Is it present in monomer or dimer outside the appropriate pH range?

2. Depending on the the presence or absence of water, the dimer and trimer transform each other. Is it possible to observe both dimer and trimer in ¹H NMR at the same time? If so, VT-NMR should be measured to determine thermodynamic parameters that will be helpful for the understanding the equilibrium between dimer and trimer.

3. For the characterization of trimer, UV-Raman was used. How about the IR spectra that was usually used for the characterization of boronic acid, boroxine, and boronic ester (Ref. Chem Mater 2014, 26, 3781.)?

4. Please provide details of experimental procedures, especially water content for reproducibility. For example, conversion from dimers to trimers, HBQ experiment, exchange reaction between different trimers, the isotope tracing experiments, fluorescence experiment, and association with F⁻, etc.

5. Please provide the summarized chemical compound data of dimer and trimer including raw spectra.

6. Please find out if it's a mistake: p.13 line 285 dimers  trimers ?

7. An overview of conventional methods of improving the stability of boroxines against hydrolysis (steric effects, electronic effects, entropically stabilizing, etc) should be added to the introduction in the main text. (Chem. Eur. J. 2023, 29, e202300995.).

Reviewer #2 (Remarks to the Author):

This manuscript entitled "Water-stable boroxine structure with dynamic covalent bonds" by Qing et al reported that the boronic acid HO-PBA exists as a dimer at ambient conditions with AIEE and dynamic exchange properties. Interestingly, HO-PBA dimer can transform into boroxine structure with the presence of water, as proved by NMR spectra, UV-Raman measurements, MS and DFT calculations. The transformed boroxine exhibits good stability, dynamic exchange properties and the ability to selectively recognize F- in aqueous. Using the derived boroxine as crosslinking points, a hydrogel with reversible gel-sol transition and high stability was synthesized. The most impressive part of this work is that a boronic acid dimer can transform into boroxine structure upon adding water, and the boroxine is very stable in aqueous media (even in acidic or alkaline solutions) but also can exchange rapidly at room temperature. These results are in contradiction to our common sense as it is well known that boroxines are formed from dehydration process and will hydrolyze into boronic acids upon exposure to water. Although the authors have tried many characterizing techniques to prove their assertions. I still think this work is not solid enough and need more evidence to support their conclusions.

1) As to the roles of water in the dimer-to-trimer reaction and its contribution to the stabilization of HO-PBA trimer, the author said that they will carry detailed reaction kinetics studies and accurate theoretical calculations to unravel the answers in the future. However, since the dimer-to-trimer reaction and the stability of HO-PBA trimer in aqueous media is so important in this work, I suggest the author to perform these studies and include the relevant results in this revised manuscript to make this work more solid.

2) The authors claimed that HO-PBA exists as a dimer in the solid state and solution, as proved by the single-crystal structure and NMR spectra. This is not a not a very rigorous expression. Actually, the HO-PBA dimer is the dehydration product of HO-PBA. They are not the same matter.

3) In Figure 2b, peak at m/z 253.20 was attributed to the exchange product between HO-PBA dimer and CH₃-HO-PBA dimer. However, the peak at m/z 253.20 is not the highest one. What is the peak at about 250?

4) The transformation from HO-PBA dimer to boroxine in aqueous was proved by ^1H and ^{13}C NMR spectra, it is necessary to further compare the chemical shift of boron atoms in ^{11}B NMR spectra of HO-PBA and boroxine.

5) To determine the boroxine structure, the authors compared the UV-Raman spectra of solid HO-PBA, $\text{CH}_3\text{-HO-PBA}$ and $\text{CF}_3\text{-HO-PBA}$ dimer samples with the corresponding spectra of the dimer samples which were dissolved in ACN-water solutions. However, for the sake of rigor, it is better to measure the UV-Raman spectra of dimers in anhydrous solvents, such as anhydrous acetonitrile.

6) The authors explored the stability of the derived boroxine structures in aqueous with wide pH ranges via fluorescence measurements. But the FL test is qualitative while not quantitative, the supplemental ^1H NMR, MS and other measurements should be carried out to further investigate the stability of boroxine in aqueous of different conditions.

7) To determine the role of water in the transformation between HO-PBA dimer and boroxine structure, the ESI-Q-TOF test was conducted, which indicated that H_2O participated the transformation process. However, the authors should further investigate and discuss the reasons for the stable existence of boroxine moieties in water.

8) ESI-Q-TOF MS measurements demonstrated that the dynamic exchanges could occur rapidly between boroxine structures in methanol-water solution. Why can the dynamic B-O bonds undergo highly dynamic fracture and recombination but still remain stable in aqueous at room temperature? Is it still stable if the boroxine structures are stored in water for a long time? Continuous detection of the structural changes through NMR, MS, Raman and other measurements should be added and discussed.

9) The authors introduced the importance of boronic acids and boroxines in the introduction. Based on the kinetic tenability and thermodynamic stability, boroxines have been utilized to construct dynamic polymers and COFs, but the advanced and classic studies about them were not introduced, such as those in 10.1002/adfm.201800560, 10.1002/adma.201602332, and 10.1021/jacs.1c05987. For the contradictory problem between stability and kinetics of boroxines, some strategies were proposed and investigated, such as the introduction of B-N coordination interactions. Some related works and methods should be introduced and discussed.

Reviewer #3 (Remarks to the Author):

The paper by Li et al. addresses a long-standing challenge in chemistry and materials science, namely the hydrolytic instability of boroxines in aqueous environments. Boroxines are versatile structures with potential applications in various fields, but their rapid degradation in water has limited their use. The discovery of a water-stable boroxine structure would open the door to a wide range of applications, including hydrogels, ion detection, repairable adhesives, biosensing, and more.

The study presents novel findings, most notably the synthesis of a water-stable boroxine structure derived from 2-hydroxyphenylboronic acid (HO-PBA). This HO-PBA trimer exhibits dynamic covalent bonding, aggregation-induced enhanced emission (AIEE), and remarkable stability over a wide pH range. It also shows selective recognition of fluoride ions in aqueous media. The results are supported by a solid

combination of experimental data, including crystallography, NMR, UV-Raman spectroscopy, and theoretical calculations. This discovery has far-reaching implications for materials science and chemistry, offering solutions to the challenges posed by boroxine hydrolysis and enabling innovative applications in aqueous environments.

The water stability of boroxine is based on two remarkable aspects. First, the remarkable tendency of HO-PBA to exist as a dimer rather than a monomer under dry conditions. Second, upon addition of liquid water, the HO-PBA dimers do not hydrolyze to monomers, but rather form HO-PBA trimers.

However, although these are the two main novelties, the authors fail to provide a proper rationale, especially for the trimer stabilization in water. While future efforts are promised by the authors, the paper in its current form cannot be considered a solid advance in the field.

A solid explanation of the role of water in stabilizing the trimer should be provided, as well as insight into the instabilities of the monomer under dry conditions. Without these, this reviewer would not recommend the paper for publication in Nature Communications.

REVIEWER COMMENTS

Reviewer #1 (Remarks to the Author):

The chemical instability of boroxines to hydrolysis is an intrinsic problem. The manuscript presented by Qing and coworkers reports an unprecedented water-stable boroxine. They found that HO-PBA exists as a dimer and interestingly the dimer transformed into a trimer (boroxine) upon exposure to water. The structure of trimer was carefully characterized and discussed by several techniques such as NMR, MS, UV-Raman, fluorescence, reactions with HBQ, Br₂ or Fe³⁺, DFT calculations, although X-ray structure of boroxine was not clarified. Furthermore, the trimer (boroxine) was found to show excellent stability against water in the pH range from 2 to 9 and unique dynamic behavior with rapid exchange between various trimers. Taking advantage of the stability and dynamic behavior, the boroxine-based dynamic hydrogel was demonstrated. These findings are very significant because they change the common knowledge about the hydrolytic instability of boroxines and represent a major breakthrough not only in the fundamental properties of boroxines but also in boroxine materials. Although the role of water in the conversion from dimer to trimer (boroxine) and the reason for the high chemical stability of trimer (boroxine) with water are not clear, the reviewer hopes these questions will be elucidated soon. The significance and novelty of this work are valuable for the scientific community. For all these reasons, I recommend the publication of this work after some minor revisions.

Response: We appreciate your positive, insightful and constructive comments on our manuscript. Inspired by your second question, we have taken a significant step towards clarifying the role of water in the conversion from dimer to trimer (boroxine) and the reason for the high chemical stability of trimer (boroxine) with water. Please refer to our response to your second question for more information. In the following response, the reviewer's comments are highlighted in blue, and our responses immediately follow.

1. Is it possible to observe a monomer (boronic acid) of HO-PBA by ¹H NMR? Can dimer or trimer be hydrolyzed completely to monomer (HO-PBA)? Is it present in monomer or dimer outside the appropriate pH range?

Response: Thanks for your question. The ¹H NMR spectra of HO-PBA trimer were examined at

different pH values. Within the pH range of 2–9, no discernible variances were observed in the ^1H NMR spectra, indicating the stability of HO-PBA trimer in this pH range (Fig.R1a). However, at pH values greater than 11, the ^1H NMR spectrum of HO-PBA trimer exhibits additional peaks (Fig.R1b), accompanied by a color change from colorless to brown in the solution. This suggests that HO-PBA trimers might undergo oxidation in strongly alkaline solutions. Consequently, strongly alkaline conditions were excluded from further investigation. When the solution's pH is below 1, the solution of HO-PBA trimer remains colorless, and the peaks in the ^1H NMR spectrum show a slight shift (Fig.R1c). Since HO-PBA trimer has a symmetric structure, its ^1H NMR spectrum is identical to that of HO-PBA monomer in terms of the number and shape of peaks. Theoretically, the peak positions of the ^1H NMR spectrum can be adopted as a criteria for differentiating between HO-PBA trimer and monomer. However, the peak position of the ^1H NMR spectrum can also change due to the solvent influence. Thus, it is challenging to determine whether the peak shifts in the ^1H NMR spectrum at the pH values below 1 are the result of the hydrolysis of HO-PBA trimer into HO-PBA monomer, or due to the varying acidic environment of the solvent.

To further investigate whether HO-PBA trimer undergoes hydrolysis to form monomer in strongly acidic solutions, we employed electrospray ionization quadrupole time-of-flight (ESI-Q-TOF) mass spectrometry. In the ESI process, the analytical solution is first pumped with a mobile phase through a capillary, and then small droplets are emitted from a Taylor cone at the tip of the ESI capillary. These droplets rapidly shrink due to solvent evaporation, ultimately leading to the formation of offspring droplets with a size of a few nm through jet fission. It is worth noting that solvent evaporation can result in extreme pH values, and even if the mildly acidic mobile phases can yield offspring droplets with a pH value of 1 or less (*Angew. Chem. Int. Ed.* 2016, 55, 2–15; *Anal. Chem.* 2001, 73, 4836–4844; *J. Am. Soc. Mass Spectrom.* 2017, 28, 1827–1835; *Anal. Chem.* 2023, 95, 13957–13966). Thus, when the mobile phase consisting of acetonitrile and 0.1% formic acid aqueous solution (1:1, v/v) was used, the offspring droplets in the ESI process provided a strongly acidic environment for HO-PBA trimer.

As shown in Fig.R1d, in the ESI-Q-TOF mass spectrum, the two strong peaks at m/z 359.13 and 373.14, corresponding to HO-PBA trimer (Fig.4a in the main text), completely disappear,

while a strong peak at m/z 137.04, corresponding to HO-PBA monomer, appears. This result indicates the complete hydrolysis of HO-PBA trimer into monomer in a strongly acidic environment.

Fig. R1 a–c ^1H NMR spectra of HO-PBA trimer at different pH values. d ESI-Q-TOF mass spectrum of HO-PBA trimer in a methanol–water (5:1, v/v) solution, acquired in a negative mode. A mixture of 0.1% formic acid aqueous solution and acetonitrile (1:1, v/v) was adopted as mobile phase.

2. Depending on the the presence or absence of water, the dimer and trimer transform each other. Is it possible to observe both dimer and trimer in 1H NMR at the same time? If so, VT-NMR should be measured to determine thermodynamic parameters that will be helpful for the understanding the equilibrium between dimer and trimer.

Response: Thanks for your constructive question. Indeed, both HO-PBA dimer and HO-PBA trimer can be observed at the same time. As shown in Fig. R2a, after adding a trace amount of D_2O (2 μL) into a solution of HO-PBA dimer in $\text{THF}-d_8$ (0.27 $\text{mol}\cdot\text{L}^{-1}$, 500 μL), the ^1H NMR spectrum exhibits coexisting NMR signals corresponding to the aromatic protons of both HO-PBA dimer (marked with red circles) and trimer (marked with blue triangles).

With an increasing amount of D₂O, the NMR signals corresponding to the aromatic protons of HO-PBA dimer gradually diminish, while signals corresponding to the aromatic protons of HO-PBA trimer emerge and gradually intensify (Fig. R2a). Eventually, the signals of HO-PBA dimer disappear, leaving only the signals of HO-PBA trimer in the ¹H NMR spectra. This indicates the complete transformation of HO-PBA dimers into trimers. Under ambient environments, upon removal of H₂O, the reaction spontaneously reverses, and HO-PBA dimers are reformed (Fig. R2b). These results demonstrate that D₂O (H₂O) acts as a reactant in the transformation process from HO-PBA dimer to trimer, and HO-PBA dimers undergo a reversible reaction with D₂O (H₂O), producing HO-PBA trimer–H₂O complexes (Fig. R2c). Water as the reactant can promote the reaction to proceed in the forward direction, thus providing a necessary environment for the stable existence of HO-PBA trimer.

Fig. R2 a ^1H NMR spectra of equilibrium mixture of HO-PBA dimer and trimer with various amount of D_2O . The initial sample was prepared by dissolving 32.8 mg HO-PBA dimer in 500 μL THF- d_8 . **b** ^1H NMR spectrum of product in THF- d_8 from lyophilization of the THF-water (1:2, v/v) solution of HO-PBA trimer at room temperature. **c** Reversible reaction between HO-PBA dimer and H_2O to produce HO-PBA trimer- H_2O complex.

Fig. R3 a VT- ^1H NMR spectra of equilibrium mixture of HO-PBA dimer and HO-PBA trimer- D_2O complex, which was prepared by dissolving 32.8 mg HO-PBA dimer in a D_2O (4.5 μL) and THF- d_8 (500 μL) mixture. **b** Van't Hoff plot of the transformation from HO-PBA dimer to HO-PBA trimer- D_2O complex.

In the case of $n=2$ (Fig. R2c), the equilibrium constants, calculated by the integrating the peak areas of HO-PBA dimer at 7.54 ppm and HO-PBA trimer–D₂O complex at 7.64 ppm (Fig. R2a), remain constant as the amount of D₂O increases (Table R1). Therefore, it can be inferred that a molecule of HO-PBA trimer binds to two molecules of D₂O (H₂O).

Furthermore, through a variable-temperature ¹H (VT-¹H) NMR study ranging from –20 °C to 25 °C (Fig. R3a), the thermodynamic parameters of the transformation from HO-PBA dimer to HO-PBA trimer–D₂O complex were determined as follows: $\Delta H = -20.10 \text{ kJ}\cdot\text{mol}^{-1}$ and $\Delta S = -6.46 \text{ J}\cdot\text{mol}^{-1}\cdot\text{K}^{-1}$ (Fig. R3b), indicating that the transformation from HO-PBA dimer to HO-PBA trimer–D₂O complex is enthalpically driven.

Table R1 Equilibrium constants for the transformation from HO-PBA dimer to HO-PBA trimer–D₂O complex, calculated at different n values and D₂O amounts under room temperature.

n	D ₂ O (μL)				
	2	4	6	8	10
1	13.2	44.6	104.9	208.5	368.3
2	2063.1	2036.5	2023.6	2041.1	2076.8
3	6.9×10^5	2.4×10^5	9.5×10^4	4.3×10^4	2.2×10^4
4	6.4×10^8	1.1×10^8	1.6×10^7	2.5×10^6	5.6×10^5

The above results have been incorporated into the part of “Role of water in the formation of the boroxine structure” in the revised manuscript.

3. For the characterization of trimer, UV-Raman was used. How about the IR spectra that was usually used for the characterization of boronic acid, boroxine, and boronic ester (Ref. Chem Mater 2014, 26, 3781.)?

Response: Thanks for your question. We have carefully read this reference, which provides characteristic IR regions for the characterization of solid-state samples of boroxine, boronic acid, and boronate ester species. However, we cannot obtain a solid HO-PBA trimer sample because of the spontaneous transformation of HO-PBA trimers to HO-PBA dimers in the absence of water.

Therefore, to maintain its stability and to enhance its solubility for meeting the concentration requirement of spectral measurement, HO-PBA trimer needs to be dissolved in a water–organic reagent mixture. Nevertheless, utilizing IR spectroscopy on solutions presents significant challenges, as solvents significantly contribute to the IR spectrum.

As shown in Fig.R4, water exhibits a broad vibrational peak in the 1000–600 cm^{-1} region (Fig.R4a), and the common organic reagents soluble in water also display abundant vibrational peaks in this region (Fig.R4b–g). These solvent peaks can severely interfere with or may even mask the characteristic vibrational peak of the boroxine structure in HO-PBA trimer (1000–600 cm^{-1}). The solvent peaks in the solution, with high intensity, differ from the pure solvent peak, making it impossible to completely eliminate these solvent peaks using spectral subtraction techniques. Therefore, IR spectroscopy is incapable of providing solid proofs for the existence of the boroxine structure in HO-PBA trimer.

Fig.R4 FT-IR spectra from the NIST Chemistry WebBook of water (a), methanol (b), acetonitrile (c), dimethyl sulfoxide (d), acetone (e), ethanol (f), and tetrahydrofuran (g).

4. Please provide details of experimental procedures, especially water content for

reproducibility. For example, conversion from dimers to trimers, HBQ experiment, exchange reaction between different trimers, the isotope tracing experiments, fluorescence experiment, and association with F⁻, etc.

Response: Thanks for your instruction. Details of experimental procedures have been provided in Supplementary Information as follows.

The fluorescence spectra of liquid samples were measured at right angles to the incident radiation, which was incident on samples in a 1 cm path-length quartz cell. For solid samples, the fluorescence spectra were measured using a solid sample holder positioned diagonally at -45 degrees with respect to the incident and output beams. In the HBQ experiment, solutions of HBQ and a mixture of HBQ and HO-PBA dimer were prepared at room temperature, with concentrations of $4.0 \times 10^{-5} \text{ mol} \cdot \text{L}^{-1}$ for HBQ and $2.0 \times 10^{-4} \text{ mol} \cdot \text{L}^{-1}$ for HO-PBA dimer in the two solutions. Water-THF mixtures in a ratio of 4:1 (v/v) were adopted as solvents. All the fluorescence spectra were recorded at room temperature with both excitation and emission slits set to 5 nm.

To investigate the selective association of the water-stable boroxine structure with F⁻, solutions of HO-PBA trimer, NaF suspension, and a mixture of HO-PBA trimer and NaF were prepared for ¹H, ¹¹B, and ¹⁹F NMR measurements. The concentrations of HO-PBA trimer and NaF were both 120 mmol·L⁻¹, and D₂O-DMSO-*d*₆ mixtures in a ratio of 1:6 (v/v) were adopted as solvents. For CH₃-HO-PBA trimer and CF₃-HO-PBA trimer, similar NMR experiments were carried out.

To investigate the exchange reaction between different dimers, the sample was prepared by co-dissolving any two of HO-PBA dimer, CH₃-HO-PBA dimer, and CF₃-HO-PBA dimer (each at a concentration of 20 mg·mL⁻¹) in dry THF at room temperature.

To examine the conversion from dimers to trimers, HO-PBA dimers (1.7 mg·mL⁻¹) were dissolved in a methanol-water (5:1, v/v) mixture at room temperature. To investigate the exchange reaction between different trimers, any two of HO-PBA trimer, CH₃-HO-PBA trimer, and CF₃-HO-PBA trimer (each at a concentration of 1.7 mg·mL⁻¹) were co-dissolved in a methanol-water (5:1, v/v) mixture at room temperature. In the isotope tracing experiment, HO-PBA dimers (1.7 mg·mL⁻¹) was dissolved in a methanol-H₂¹⁸O (5:1, v/v) mixture at room temperature. All samples were

filtered using 0.22 μm pore-size membrane filters.

5. Please provide the summarized chemical compound data of dimer and trimer including raw spectra.

Response: The summarized chemical compound data of dimer and trimer, including the single crystal data, raw NMR spectra, raw Raman spectra and mass spectra, have been upload into “Supplementary Dataset” in the submission system.

6. Please find out if it's a mistake: p.13 line 285 dimers  trimers ?

Response: Thanks for your correction. The word “dimers” has been changed into trimers.

7. An overview of conventional methods of improving the stability of boroxines against hydrolysis (steric effects, electronic effects, entropically stabilizing, etc) should be added to the introduction in the main text. (Chem. Eur. J. 2023, 29, e202300995.).

Response: Thanks for your instruction. An overview of conventional methods for improving the stability of boroxines against hydrolysis has been added to the introduction in the revised manuscript as follows.

“Efforts to enhance the stability of boroxines against hydrolysis have traditionally focused on reducing the electrophilicity of the Lewis acidic boron sites. Strategies include introducing electron-donating groups, incorporating bulky groups, and forming adducts with N-donor ligands¹⁸⁻²⁰. Recently, Ono, *et al* proposed an entropic stabilization strategy by incorporating three boronic acid units into a flexible macrocycle²⁰. Although these methods have made boroxines more robust against hydrolysis, they have not fundamentally addressed the underlying issue of hydrolytic instability of boroxines.”

Related references have been added to the revised manuscript.

18. Bapat, A. P., Sumerlin, B. S. & Sutti, A. Bulk network polymers with dynamic B–O bonds: healable and reprocessable materials. *Mater. Horizons* 7, 694–714 (2020).

19. Korich, A. L. & Iovine, P. M. Boroxine chemistry and applications: A perspective. *Dalton T.*

39, 1423–1431, (2010).

20. Ono, K., Sawanaga, K., Onodera, S., Kawai, H. & Goto, K. Structural interconversion based on intramolecular boroxine formation. *Chem. Eur. J.* **29**, e202300995, (2023).

Reviewer #2 (Remarks to the Author):

This manuscript entitled “Water-stable boroxine structure with dynamic covalent bonds” by Qing et al reported that the boronic acid HO-PBA exists as a dimer at ambient conditions with AIEE and dynamic exchange properties. Interestingly, HO-PBA dimer can transform into boroxine structure with the presence of water, as proved by NMR spectra, UV-Raman measurements, MS and DFT calculations. The transformed boroxine exhibits good stability, dynamic exchange properties and the ability to selectively recognize F⁻ in aqueous. Using the derived boroxine as crosslinking points, a hydrogel with reversible gel-sol transition and high stability was synthesized. The most impressive part of this work is that a boronic acid dimer can transform into boroxine structure upon adding water, and the boroxine is very stable in aqueous media (even in acidic or alkaline solutions) but also can exchange rapidly at room temperature. These results are in contradiction to our common sense as it is well known that boroxines are formed from dehydration process and will hydrolyze into boronic acids upon exposure to water. Although the authors have tried many characterizing techniques to prove their assertions. I still think this work is not solid enough and need more evidence to support their conclusions.

Response: Thank you very much for taking time to review our manuscript, and your thoughtful insights and comments are invaluable for improving the quality of this work. According to your comments, we have conducted additional experiments and made corrections to address the shortcomings of this work. In the following response, the reviewer’s comments are highlighted in blue, and our responses immediately follow.

1) As to the roles of water in the dimer-to-trimer reaction and its contribution to the stabilization of HO-PBA trimer, the author said that they will carry detailed reaction kinetics studies and accurate theoretical calculations to unravel the answers in the future. However, since the dimer-to-trimer reaction and the stability of HO-PBA trimer in aqueous media is so important in this work, I suggest the author to perform these studies and include the relevant results in this revised manuscript to make this work more solid.

7) To determine the role of water in the transformation between HO-PBA dimer and boroxine structure, the ESI-Q-TOF test was conducted, which indicated that H₂O participated the

transformation process. However, the authors should further investigate and discuss the reasons for the stable existence of boroxine moieties in water.

Response: Thank you for your constructive suggestion and guidance. Given that both the first and the seventh questions concern about the same issue, we have combined them to provide a unified response. Our studies have focused on investigating the roles of water in the dimer-to-trimer reaction and its contribution to the stabilization of HO-PBA trimer. The relevant results have been incorporated into the part of “Role of water in the formation of the boroxine structure” in the revised manuscript as follows.

Fig. S1 a ¹H NMR spectra of equilibrium mixture of HO-PBA dimer and trimer with various amount of D₂O. The initial sample was prepared by dissolving 32.8 mg HO-PBA dimer in 500 μL THF-*d*₈. **b** ¹H NMR spectrum of product in THF-*d*₈ from lyophilization of the THF-water (1:2, v/v) solution

of HO-PBA trimer at room temperature. **c** Reversible reaction between HO-PBA dimer and H₂O to produce HO-PBA trimer–H₂O complex.

The involvement of H₂O in the transformation process of HO-PBA dimer to trimer, along with the shared empirical formula between the two species, suggests a potential catalytic function of H₂O in this process. To validate this hypothesis, we conducted a NMR titration experiment by gradually introducing D₂O into a solution of HO-PBA dimer in THF-*d*₈. With increasing amounts of D₂O, the NMR signals corresponding to the aromatic protons of HO-PBA dimer (marked with red circles) gradually diminish. Simultaneously, signals corresponding to the aromatic protons of HO-PBA trimer (marked with blue triangles) emerge and gradually intensify (Fig.S1a). Eventually, the signals of HO-PBA dimer disappear, leaving only the signals of HO-PBA trimer in the ¹H NMR spectra. This indicates the complete conversion of HO-PBA dimers into trimers. Under ambient environments, upon removal of H₂O, the reaction spontaneously proceeds in the opposite direction, resulting in the regeneration of HO-PBA dimers (Fig.S1b). These results demonstrate that D₂O (or H₂O) acts as a reactant rather than a catalyst in the transformation process from HO-PBA dimer to trimer (Fig.S1c), and HO-PBA dimers undergo a reversible reaction with D₂O (or H₂O), producing HO-PBA trimer–H₂O complexes (simplified as HO-PBA trimer for convenience).

The transformation from HO-PBA dimer to trimer was further investigated using ²D NMR spectroscopy. In the ²D NMR spectrum of THF, no peaks are observed due to the extremely low deuterium content of THF. After adding a trace amount of D₂O (2 μL) into THF (500 μL), a distinct peak attributed to D₂O appears at 2.4 ppm. With increasing amounts of D₂O, this peak progressively intensifies and shifts towards lower field (Fig.S2a). Interestingly, after adding a trace amount of D₂O (2 μL) into a THF solution of HO-PBA dimer (0.27 mol·L⁻¹, 500 μL), no peak at 2.4 ppm emerges. Instead, three peaks labeled as n, m, and x appear in the range of 7–9.5 ppm (Fig.S2b), indicating the presence of three types of active hydrogen in the solution. The peak “m” is assigned to the deuterium of the boron hydroxyl group in HO-PBA dimer, which gradually disappears with increasing amounts of D₂O due to the transformation of HO-PBA dimer to trimer. The peak “n” is attributed to the deuterium of phenolic hydroxyl groups in both HO-PBA dimer and trimer, and they could not be discriminated due to their similar chemical shifts. The peak “x” should be assigned to D₂O in HO-PBA trimer–D₂O complex, with its chemical shift changing from 2.4 to 7.5 ppm as a

result of the binding interaction with HO-PBA trimer. Furthermore, upon addition of more D₂O, the binding sites of D₂O in HO-PBA trimer become saturated. Consequently, a signal corresponding to free D₂O emerges at 3.3 ppm.

Fig. S2 a ²D NMR spectra of THF (500 μL) with various amount of D₂O. b ²D NMR spectra of

equilibrium mixture of HO-PBA dimer and trimer with various amount of D₂O. The initial sample was prepared by dissolving 32.8 mg HO-PBA dimer in 500 μL THF. **c** ¹¹B NMR spectra of HO-PBA dimer (black line) in THF-*d*₈ and HO-PBA trimer (red line) in D₂O-THF-*d*₈ (1:5, v/v) mixture, which were determined relative to 0.1 mol·L⁻¹ boric acid as the external reference (19.4 ppm, marked with a star). **d** ²D NMR spectra of PBA (0.27 mol·L⁻¹) and phenol (0.27 mol·L⁻¹) in a THF-D₂O (25:1, v/v) solution.

Subsequently, to investigate the binding site of D₂O, ¹¹B NMR spectroscopy was adopted, which is sensitive to the coordination environment of boron atoms. If the binding site of D₂O was on the boron atom, the chemical shift of HO-PBA trimer would move to higher field compared to that of HO-PBA dimer. However, it exhibits a light shift towards lower field, indicating that the binding site of D₂O is not on the boron atom (Fig. S2c). Therefore, D₂O might bind to HO-PBA trimer through hydrogen bond formed between the deuterium atoms of D₂O and oxygen atoms of the boroxine structure in HO-PBA trimer.

In the ²D NMR spectrum of phenylboronic acid (PBA) and phenol in the THF-D₂O solution, no signals corresponding to the bound D₂O are observed (Fig. S2d), indicating the non-formation of PBA-D₂O and phenol-D₂O complex. Therefore, the capability to form a complex by binding with D₂O is a distinctive characteristic of the boroxine structure present in HO-PBA trimer.

Last, the number of D₂O bound to a molecule of HO-PBA trimer was determined. In the case of *n*=2 (Fig. S1c), the equilibrium constants, calculated by the integrating the peak areas of HO-PBA dimer at 7.54 ppm and HO-PBA trimer-D₂O complex at 7.64 ppm (Fig. S1a), remain constant as the amount of D₂O increases (Table S1). Based on this, it can be inferred that a HO-PBA trimer binds to two molecules of D₂O. Furthermore, through a variable-temperature ¹H (VT-¹H) NMR study ranging from -20 °C to 25 °C (Fig. S3a), the thermodynamic parameters of the transformation from HO-PBA dimer to HO-PBA trimer-D₂O complex were determined as follows: Δ*H*=-20.10 kJ·mol⁻¹ and Δ*S*=-6.46 J·mol⁻¹·K⁻¹ (Fig.S3b), indicating that the transformation from HO-PBA dimer to HO-PBA trimer-D₂O complex is enthalpically driven.

Table S1 Equilibrium constants for the transformation from HO-PBA dimer to HO-PBA trimer-D₂O complex, calculated at different *n* values and D₂O amounts under room temperature.

n	D ₂ O (μL)				
	2	4	6	8	10
1	13.2	44.6	104.9	208.5	368.3
2	2063.1	2036.5	2023.6	2041.1	2076.8
3	6.9×10 ⁵	2.4×10 ⁵	9.5×10 ⁴	4.3×10 ⁴	2.2×10 ⁴
4	6.4×10 ⁸	1.1×10 ⁸	1.6×10 ⁷	2.5×10 ⁶	5.6×10 ⁵

Fig. S3 a VT-¹H NMR spectra of equilibrium mixture of HO-PBA dimer and HO-PBA trimer–D₂O complex, which was prepared by dissolving 32.8 mg HO-PBA dimer in a D₂O (4.5 μL) and THF–*d*₈ (500 μL) mixture. **b** Van't Hoff plot of the transformation from HO-PBA dimer to HO-PBA trimer–D₂O complex.

2) The authors claimed that HO-PBA exists as a dimer in the solid state and solution, as proved by the single-crystal structure and NMR spectra. This is not a not a very rigorous expression.

Actually, the HO-PBA dimer is the dehydration product of HO-PBA. They are not the same matter.

Response: Thanks for your instruction. The relevant expression, e.g., we find that HO-PBA exists as a unique dimer, rather than a monomer as stated in the literature, has been revised to “we find that, under ambient environments, HO-PBA undergoes spontaneous dehydration to form a dimer, which challenges the existing literature's claim of its existence solely as a monomer” in the revised manuscript.

3) In Figure 2b, peak at m/z 253.20 was attributed to the exchange product between HO-PBA dimer and CH₃-HO-PBA dimer. However, the peak at m/z 253.20 is not the highest one. What is the peak at about 250?

Response: Thanks for your question. We individually analyzed the MALDI-TOF mass spectra of HO-PBA dimer (black line) and CH₃-HO-PBA dimer (red line) (Fig. S4). In addition to the peaks at m/z 239.18 and 267.22, which correspond to HO-PBA dimer and CH₃-HO-PBA dimer respectively, both spectra also display a prominent peak at m/z 250.24. Therefore, we presumed that this peak could potentially be attributed to a background signal, which we have incorporated into the caption of Fig. 2b in the revised manuscript.

Fig.S4 MALDI-TOF mass spectrum of HO-PBA dimers (black line), CH₃-HO-PBA dimers (red line), and products from the mixture of HO-PBA dimers and CH₃-HO-PBA dimers (green line), acquired in a negative mode.

4) The transformation from HO-PBA dimer to boroxine in aqueous was proved by ¹H and ¹³C NMR spectra, it is necessary to further compare the chemical shift of boron atoms in ¹¹B NMR spectra of HO-PBA and boroxine.

Response: Thanks for your guidance. The ¹¹B NMR spectra of HO-PBA dimer and trimer were measured (Fig.S5) and incorporated as Fig. 5c in the revised manuscript. The chemical shifts of ¹¹B were determined relative to 0.1 mol·L⁻¹ boric acid (19.4 ppm, marked with a star), which served as the external reference (*Electroanalysis* 2010, 22, 1337–1343). As shown in Fig.S5, the chemical shift of ¹¹B in HO-PBA trimer exhibits a light shift towards lower field compared to that in HO-PBA dimer.

Fig.S5 ¹¹B NMR spectra of HO-PBA dimer (black line) in THF-*d*₈ and HO-PBA trimer (red line) in a D₂O-THF-*d*₈ (1:5, v/v) mixture.

5) To determine the boroxine structure, the authors compared the UV-Raman spectra of solid HO-PBA, CH₃-HO-PBA and CF₃-HO-PBA dimer samples with the corresponding spectra of the dimer samples which were dissolved in ACN-water solutions. However, for the sake of rigor, it is better to measure the UV-Raman spectra of dimers in anhydrous solvents, such as anhydrous acetonitrile.

Response: Thanks for your guidance. The UV-Raman spectra of dimers in anhydrous acetonitrile

were measured (Fig.S6). Specifically, Fig.S6a was incorporated as Fig. 4d in the revised manuscript, while Fig.S6b and S6c were incorporated as Supplementary Fig. 13g and 17b, respectively, in the revised Supplementary Information. As shown in Fig. S6a, the UV–Raman spectrum of HO-PBA trimer (green line) is significantly different from that of HO-PBA dimer in the solid state (black line) and in a dry acetonitrile (ACN) solution (red line). It exhibits four characteristic peaks of the boroxine ring at 1032, 835, 705, and 579 cm^{-1} , respectively, along with a peak at 922 cm^{-1} assigned to the C-C stretching of ACN. Similar spectral features were also observed in Fig.S6b and S6c. The discrepancies in peak positions among HO-PBA trimer, CH_3 -HO-PBA trimer, and CF_3 -HO-PBA trimer could be attributed to the influence of the substituents on the skeletal vibration of CH_3 -HO-PBA trimer and CF_3 -HO-PBA trimer.

Fig.S6 UV–Raman spectra of the solid sample (black), sample in a dry ACN solution (red) and sample in an ACN–water (1:2, v/v) solution (green).

6) The authors explored the stability of the derived boroxine structures in aqueous with wide pH ranges via fluorescence measurements. But the FL test is qualitative while not quantitative,

the supplemental ^1H NMR, MS and other measurements should be carried out to further investigate the stability of boroxine in aqueous of different conditions.

Response: Thanks for your guidance. In the MS measurement, the solution pH might be altered during the analytical workflow (*Angew. Chem. Int. Ed.* 2016, 55, 2–15; *Anal. Chem.* 2001, 73, 4836–4844; *J. Am. Soc. Mass Spectrom.* 2017, 28, 1827–1835; *Anal. Chem.* 2023, 95, 13957–13966), rendering MS unsuitable for exploring the stability of boroxine structures at various pH values. Therefore, ^1H NMR and UV–Raman measurements were carried out for this study. As shown in Fig.S7 (incorporated as supplementary Fig. 19 in the revised Supplementary Information), no variations were observed in the ^1H NMR and UV–Raman spectra of HO-PBA trimer and $\text{CH}_3\text{-HO-PBA}$ trimer, indicating the stability of the boroxine structure across a wide pH range.

Fig.S7 a, c ^1H NMR spectra of HO-PBA trimer (**a**) and $\text{CH}_3\text{-HO-PBA}$ trimer (**c**) at different pH values in a D_2O –ethanol- d_6 (1:1, v/v) solution. **b, d** UV–Raman spectra of HO-PBA trimer (**b**) and $\text{CH}_3\text{-HO-PBA}$ trimer (**d**) at different pH values in an ACN–water (1:2, v/v) solution.

8) **ESI-Q-TOF MS measurements demonstrated that the dynamic exchanges could occur rapidly between boroxine structures in methanol-water solution. Why can the dynamic B-O**

bonds undergo highly dynamic fracture and recombination but still remain stable in aqueous at room temperature? Is it still stable if the boroxine structures are stored in water for a long time? Continuous detection of the structural changes through NMR, MS, Raman and other measurements should be added and discussed.

Response: Thanks for your questions and guidance. The highly dynamic exchange of B-O bonds in boroxine and the stability of boroxine are not contradictory aspects, but belong to dynamics and thermodynamics, respectively (Fig.S8). The highly dynamic exchange of B-O bonds in boroxine suggests a low energy barrier for this process, and its transition can be propelled by the molecular thermal motion at room temperature. The free energy (ΔG_1) for the dehydration of HO-PBA to HO-PBA dimer was calculated to be $-21.4 \text{ kJ}\cdot\text{mol}^{-1}$ at 25°C through density functional theory method with the M06-2X functional and TZVP basis sets (Table S2). As stated in the response to the first question, the thermodynamic parameters of the transformation from HO-PBA dimer to HO-PBA trimer-D₂O complex were determined as follows: $\Delta H=-20.10 \text{ kJ}\cdot\text{mol}^{-1}$ and $\Delta S=-6.46 \text{ J}\cdot\text{mol}^{-1}\cdot\text{K}^{-1}$. Utilizing these values, we calculated that the free energy (ΔG_2) of the transformation from HO-PBA dimer to trimer-D₂O complex is $-18.17 \text{ kJ}\cdot\text{mol}^{-1}$ at 25°C . The low energy of HO-PBA trimer-D₂O complex contributes to its stability.

Table S2 Energy change for the dehydration of HO-PBA monomer to HO-PBA dimer, calculated by density functional theory with M06-2X functional and TZVP basis sets

	HO-PBA monomer (A)	HO-PBA dimer (B)	H ₂ O	
Substances				$2A \rightarrow B + 2H_2O$
Gibbs Free Energy ($\text{kJ}\cdot\text{mol}^{-1}$)	-1269134.9	-2137012.4	-200639.4	-21.4

Fig.S8 Schematic of the energy level diagram of HO-PBA trimer–H₂O complex, HO-PBA dimer and HO-PBA monomer.

Continuous monitoring of the structural changes of HO-PBA, CH₃-HO-PBA and CF₃-HO-PBA trimers were conducted using ¹H NMR, MS and Raman techniques. The results suggest HO-PBA, CH₃-HO-PBA and CF₃-HO-PBA trimers remain stable under ambient environments during the 7-day study (Fig.S9, incorporated as supplementary Fig. 12 in the revised Supplementary Information).

Fig.S9 a–c Comparison of ¹H NMR spectra of HO-PBA trimer (a), CH₃-HO-PBA trimer (b), or CF₃-HO-PBA trimer (c) in a D₂O–DMSO-*d*₆ (2:1, v/v) solution on the 1st and 7th day. **d–f**

Comparison of UV–Raman spectra of HO-PBA trimer (**d**), CH₃-HO-PBA trimer (**e**), or CF₃-HO-PBA trimer (**f**) in an ACN–water (1:2, v/v) solution on the 1st and 7th day. **g–i** Comparison of ESI-Q-TOF mass spectra of HO-PBA trimer (**g**), CH₃-HO-PBA trimer (**h**), or CF₃-HO-PBA trimer (**i**) in a methanol–water (5:1, v/v) solution on the 1st and 7th day, acquired in a negative mode.

9) **The authors introduced the importance of boronic acids and boroxines in the introduction. Based on the kinetic tenability and thermodynamic stability, boroxines have been utilized to construct dynamic polymers and COFs, but the advanced and classic studies about them were not introduced, such as those in 10.1002/adfm.201800560, 10.1002/adma.201602332, and 10.1021/jacs.1c05987. For the contradictory problem between stability and kinetics of boroxines, some strategies were proposed and investigated, such as the introduction of B-N coordination interactions. Some related works and methods should be introduced and discussed.**

Response: Thanks for your guidance. In the revised manuscript, the three classic studies have been included as references 11, 12, and 15.

“Furthermore, the dynamic nature of boron–oxygen (B–O) bonds in boroxines, with the unique combination of kinetic tenability and high thermodynamic stability, has rendered them as promising dynamic covalent motifs for the development of malleable and healable polymers^{5,10,11,12}, and this property has paved the way for covalent organic frameworks (COFs)¹³⁻¹⁵.”

The revised manuscript introduces and discusses related works and methods aimed at enhancing the stability of boroxines against hydrolysis.

“Efforts to enhance the stability of boroxines against hydrolysis have traditionally focused on reducing the electrophilicity of the Lewis acidic boron sites. Strategies include introducing electron-donating groups, incorporating bulky groups, and forming adducts with N-donor ligands¹⁸⁻²⁰. Recently, Ono, *et al* proposed an entropic stabilization strategy by incorporating three boronic acid units into a flexible macrocycle²⁰. Although these methods have made boroxines more robust against hydrolysis, they have not fundamentally addressed the underlying issue of hydrolytic instability of boroxines.”

References:

11. Bao, C., Jiang, Y.-J., Zhang, H., Lu, X. & Sun, J. Room-temperature self-healing and recyclable tough polymer composites using nitrogen-coordinated boroxines. *Adv. Funct. Mater.* **28**, 1800560 (2018).
12. Lai, J.-C., Mei, J.-F., Jia, X.-Y., Li, C.-H., You, X.-Z. & Bao, Z. A stiff and healable polymer based on dynamic-covalent boroxine bonds. *Adv. Mater.* **28**, 8277–8282 (2016).
15. Hamzehpoor, E., Jonderian, A., McCalla, E. & Perepichka, D. F. Synthesis of boroxine and dioxaborole covalent organic frameworks via transesterification and metathesis of pinacol boronates. *J. Am. Chem. Soc.* **143**, 13274–13280 (2021).
18. Bapat, A. P., Sumerlin, B. S. & Sutti, A. Bulk network polymers with dynamic B–O bonds: healable and reprocessable materials. *Mater. Horizons* **7**, 694–714 (2020).
19. Korich, A. L. & Iovine, P. M. Boroxine chemistry and applications: A perspective. *Dalton T.* **39**, 1423–1431, (2010).
20. Ono, K., Sawanaga, K., Onodera, S., Kawai, H. & Goto, K. Structural interconversion based on intramolecular boroxine formation. *Chem. Eur. J.* **29**, e202300995, (2023).

Reviewer #3 (Remarks to the Author):

The paper by Li et al. addresses a long-standing challenge in chemistry and materials science, namely the hydrolytic instability of boroxines in aqueous environments. Boroxines are versatile structures with potential applications in various fields, but their rapid degradation in water has limited their use. The discovery of a water-stable boroxine structure would open the door to a wide range of applications, including hydrogels, ion detection, repairable adhesives, biosensing, and more.

The study presents novel findings, most notably the synthesis of a water-stable boroxine structure derived from 2-hydroxyphenylboronic acid (HO-PBA). This HO-PBA trimer exhibits dynamic covalent bonding, aggregation-induced enhanced emission (AIEE), and remarkable stability over a wide pH range. It also shows selective recognition of fluoride ions in aqueous media. The results are supported by a solid combination of experimental data, including crystallography, NMR, UV-Raman spectroscopy, and theoretical calculations. This discovery has far-reaching implications for materials science and chemistry, offering solutions to the challenges posed by boroxine hydrolysis and enabling innovative applications in aqueous environments.

The water stability of boroxine is based on two remarkable aspects. First, the remarkable tendency of HO-PBA to exist as a dimer rather than a monomer under dry conditions. Second, upon addition of liquid water, the HO-PBA dimers do not hydrolyze to monomers, but rather form HO-PBA trimers.

However, although these are the two main novelties, the authors fail to provide a proper rationale, especially for the trimer stabilization in water. While future efforts are promised by the authors, the paper in its current form cannot be considered a solid advance in the field.

A solid explanation of the role of water in stabilizing the trimer should be provided, as well as insight into the instabilities of the monomer under dry conditions. Without these, this reviewer would not recommend the paper for publication in Nature Communications.

Response: Thank you very much for your positive and encouraging comments on this manuscript. We completely agree with your perspective regarding the significance of providing insight into the

instabilities of the monomer under dry conditions and the role of water in stabilizing the trimer. We have made a great effort to address these issues, aiming to provide an in-depth understanding.

Firstly, the structure of the HO-PBA chemical (Aladdin Reagent, Shanghai, Product No. H101964) was further characterized by the solid-state ^{13}C NMR spectroscopy, besides the previously employed liquid-state ^{13}C NMR spectroscopy. As shown in Fig.T1, the solid-state ^{13}C NMR spectrum of HO-PBA chemical is in good accordance with the liquid-state ^{13}C NMR spectrum of HO-PBA. They display all the signals of HO-PBA dimer, except for the signals of the two C atoms bonded to B atoms (marked with 1 and 7 in the molecular structure of HO-PBA dimer) due to the quadrupolar relaxation mechanism of ^{11}B nucleus. These results demonstrate that, under ambient environments, HO-PBA undergoes spontaneous dehydration to form a dimer. Additionally, the solid-state ^{13}C NMR spectra of $\text{CH}_3\text{-HO-PBA}$ chemical (Bide Pharmatech, Shanghai, Product No. BD217886), and $\text{CF}_3\text{-HO-PBA}$ chemical (Leyan, Shanghai, Product No. 1201389) also further support their dimeric structures (Fig.T2). These results further suggest that the formation of dimeric structures might be a common characteristic of HO-PBA and its derivatives.

Fig.T1 **a** ^{13}C NMR spectrum of HO-PBA in $\text{THF-}d_8$. **b** Solid-state ^{13}C NMR spectrum of the HO-PBA chemical (Aladdin Reagent, Shanghai, Product No. H101964).

Fig.T2 a, b Solid-state ^{13}C NMR spectrum of $\text{CH}_3\text{-HO-PBA}$ chemical (**a**, Bide Pharmatech, Shanghai, Product No. BD217886), and $\text{CF}_3\text{-HO-PBA}$ chemical (**b**, Leyan, Shanghai, Product No. 1201389).

To make clear the reason behind the instability of HO-PBA monomer under ambient environments, density functional theory (DFT) calculations were performed using the M06-2X functional and TZVP basis sets. The free energy for the dehydration of HO-PBA monomer to HO-PBA dimer was calculated to be $-21.4 \text{ kJ}\cdot\text{mol}^{-1}$ (Table T1), indicating that the instability of HO-PBA monomer under ambient environments can be attributed to the inherent thermodynamic preference for dimerization.

Table T1 Energy change for the dehydration of HO-PBA monomer to HO-PBA dimer, calculated by density functional theory with M06-2X functional and TZVP basis sets

	HO-PBA monomer (A)	HO-PBA dimer (B)	H_2O	
Substances				$2\text{A} \rightarrow \text{B} + 2\text{H}_2\text{O}$
Gibbs Free Energy ($\text{kJ}\cdot\text{mol}^{-1}$)	-1269134.9	-2137012.4	-200639.4	-21.4

The above results have been incorporated into the part of “HO-PBA dimer” in the revised manuscript.

Secondly, We have conducted studies to investigate the roles of water in the dimer-to-trimer reaction and its contribution to the stabilization of HO-PBA trimer. The relevant results have been incorporated into the part of “Role of water in the formation of the boroxine structure” in the revised manuscript as follows.

Fig. T3 a ^1H NMR spectra of equilibrium mixture of HO-PBA dimer and trimer with various amount of D_2O . The initial sample was prepared by dissolving 32.8 mg HO-PBA dimer in 500 μL $\text{THF-}d_8$. **b** ^1H NMR spectrum of product in $\text{THF-}d_8$ from lyophilization of the THF -water (1:2, v/v) solution of HO-PBA trimer at room temperature. **c** Reversible reaction between HO-PBA dimer and H_2O to produce HO-PBA trimer- H_2O complex.

The involvement of H₂O in the transformation process of HO-PBA dimer to trimer, along with the shared empirical formula between the two species, suggests a potential catalytic function of H₂O in this process. To validate this hypothesis, we conducted a NMR titration experiment by gradually introducing D₂O into a solution of HO-PBA dimer in THF-*d*₈. With increasing amounts of D₂O, the NMR signals corresponding to the aromatic protons of HO-PBA dimer (marked with red circles) gradually diminish. Simultaneously, signals corresponding to the aromatic protons of HO-PBA trimer (marked with blue triangles) emerge and gradually intensify (Fig.T3a). Eventually, the signals of HO-PBA dimer disappear, leaving only the signals of HO-PBA trimer in the ¹H NMR spectra. This indicates the complete conversion of HO-PBA dimers into trimers. Under ambient environments, upon removal of H₂O, the reaction spontaneously proceeds in the opposite direction, resulting in the regeneration of HO-PBA dimers (Fig.T3b). These results demonstrate that D₂O (or H₂O) acts as a reactant rather than a catalyst in the transformation process from HO-PBA dimer to trimer (Fig.T3c), and HO-PBA dimers undergo a reversible reaction with D₂O (or H₂O), producing HO-PBA trimer–H₂O complexes (simplified as HO-PBA trimer for convenience).

The transformation from HO-PBA dimer to trimer was further investigated using ²D NMR spectroscopy. In the ²D NMR spectrum of THF, no peaks are observed due to the extremely low deuterium content of THF. After adding a trace amount of D₂O (2 μL) into THF (500 μL), a distinct peak attributed to D₂O appears at 2.4 ppm. With increasing amounts of D₂O, this peak progressively intensifies and shifts towards lower field (Fig.T4a). Interestingly, after adding a trace amount of D₂O (2 μL) into a THF solution of HO-PBA dimer (0.27 mol·L⁻¹, 500 μL), no peak at 2.4 ppm emerges. Instead, three peaks labeled as n, m, and x appear in the range of 7–9.5 ppm (Fig.T4b), indicating the presence of three types of active hydrogen in the solution. The peak “m” is assigned to the deuterium of the boron hydroxyl group in HO-PBA dimer, which gradually disappears with increasing amounts of D₂O due to the transformation of HO-PBA dimer to trimer. The peak “n” is attributed to the deuterium of phenolic hydroxyl groups in both HO-PBA dimer and trimer, and they could not be discriminated due to their similar chemical shifts. The peak “x” should be assigned to D₂O in HO-PBA trimer–D₂O complex, with its chemical shift changing from 2.4 to 7.5 ppm as a result of the binding interaction with HO-PBA trimer. Furthermore, upon addition of more D₂O, the binding sites of D₂O in HO-PBA trimer become saturated. Consequently, a signal corresponding to

free D₂O emerges at 3.3 ppm.

Fig. T4 **a** ^2D NMR spectra of THF (500 μL) with various amount of D_2O . **b** ^2D NMR spectra of equilibrium mixture of HO-PBA dimer and trimer with various amount of D_2O . The initial sample was prepared by dissolving 32.8 mg HO-PBA dimer in 500 μL THF. **c** ^{11}B NMR spectra of HO-PBA dimer (black line) in $\text{THF}-d_8$ and HO-PBA trimer (red line) in $\text{D}_2\text{O}-\text{THF}-d_8$ (1:5, v/v) mixture, which were determined relative to 0.1 mol/L boric acid as the external reference (19.4 ppm, marked with a star). **d** ^2D NMR spectra of PBA ($0.27 \text{ mol}\cdot\text{L}^{-1}$) and phenol ($0.27 \text{ mol}\cdot\text{L}^{-1}$) in a $\text{THF}-\text{D}_2\text{O}$ (25:1, v/v) solution.

Subsequently, to investigate the binding site of D_2O , ^{11}B NMR spectroscopy was adopted, which is sensitive to the coordination environment of boron atoms. If the binding site of D_2O was on the boron atom, the chemical shift of HO-PBA trimer would move to higher field compared to that of HO-PBA dimer. However, it exhibits a light shift towards lower field, indicating that the binding site of D_2O is not on the boron atom (Fig. T4c). Therefore, D_2O might bind to HO-PBA trimer through hydrogen bond formed between the deuterium atoms of D_2O and oxygen atoms of the boroxine structure in HO-PBA trimer.

In the ^2D NMR spectrum of phenylboronic acid (PBA) and phenol in the $\text{THF}-\text{D}_2\text{O}$ solution, no signals corresponding to the bound D_2O are observed (Fig. T4d), indicating the non-formation of PBA- D_2O and phenol- D_2O complex. Therefore, the capability to form a complex by binding with D_2O is a distinctive characteristic of the boroxine structure present in HO-PBA trimer.

Last, the number of D_2O bound to a molecule of HO-PBA trimer was determined. In the case of $n=2$ (Fig. T3c), the equilibrium constants, calculated by the integrating the peak areas of HO-PBA dimer at 7.54 ppm and HO-PBA trimer- D_2O complex at 7.64 ppm (Fig. T3a), remain constant as the amount of D_2O increases (Table T2). Based on this, it can be inferred that a HO-PBA trimer binds to two molecules of D_2O . Furthermore, through a variable-temperature ^1H (VT- ^1H) NMR study ranging from $-20\text{ }^\circ\text{C}$ to $25\text{ }^\circ\text{C}$ (Fig. T5a), the thermodynamic parameters of the transformation from HO-PBA dimer to HO-PBA trimer- D_2O complex were determined as follows: $\Delta\text{H}=-20.10 \text{ kJ}\cdot\text{mol}^{-1}$ and $\Delta\text{S}=-6.46 \text{ J}\cdot\text{mol}^{-1}\cdot\text{K}^{-1}$ (Fig. T5b), indicating that the transformation from HO-PBA dimer to HO-PBA trimer- D_2O complex is enthalpically driven.

Table T2 Equilibrium constants for the transformation from HO-PBA dimer to HO-PBA trimer-

D₂O complex, calculated at different n values and D₂O amounts under room temperature.

n	D ₂ O (μL)				
	2	4	6	8	10
1	13.2	44.6	104.9	208.5	368.3
2	2063.1	2036.5	2023.6	2041.1	2076.8
3	6.9×10 ⁵	2.4×10 ⁵	9.5×10 ⁴	4.3×10 ⁴	2.2×10 ⁴
4	6.4×10 ⁸	1.1×10 ⁸	1.6×10 ⁷	2.5×10 ⁶	5.6×10 ⁵

Fig. T5 a VT-¹H NMR spectra of equilibrium mixture of HO-PBA dimer and HO-PBA trimer–D₂O complex, which was prepared by dissolving 32.8 mg HO-PBA dimer in a D₂O (4.5 μL) and THF–*d*₈ (500 μL) mixture. **b** Van't Hoff plot of the transformation from HO-PBA dimer to HO-PBA trimer–D₂O complex.

Thank you once again for your insightful comments. While we have endeavored to address the

issues regarding the instabilities of the monomer under dry conditions and the role of water in stabilizing the trimer, the current results may not provide an overall answer. If you have any further suggestions, we are willing to try. We will continue to advance our research in this area.

REVIEWERS' COMMENTS

Reviewer #1 (Remarks to the Author):

The authors have addressed the comments and the manuscript is quite better with additional experiments, including one on the role of water. I think it can be published in the journal as is.

Reviewer #2 (Remarks to the Author):

The authors have tried their best to improve the manuscript according to the reviewers' comments. Most of the reviewers' concerns have been addressed. Although more evidences are still necessary to fully elucidate the role of water in the conversion from dimer to trimer and the reason for the high chemical stability of trimer, I believe most of the conclusions in this manuscript have been well-supported by the the present data. Therefore, I recommend the publication of this work without further revision.

Reviewer #3 (Remarks to the Author):

The authors have addressed all the criticisms raised. Therefore, I would recommend the paper for publication in Nature Communications as it stands.